

**Simultaneous shifts in stoichiometric and fatty acid composition**
**of *Emiliania huxleyi* in response to environmental changes**
**Rong Bi[1,2], Stefanie M. H. Ismar[2], Ulrich Sommer[2] and Meixun Zhao[1]**
[1]Key Laboratory of Marine Chemistry Theory and Technology, Ocean University of
China, Ministry of Education/Laboratory for Marine Ecology and Environmental
Science, Qingdao National Laboratory for Marine Science and Technology, Qingdao,
266000, China
[2]Marine Ecology, GEOMAR Helmholtz-Zentrum für Ozeanforschung, Kiel, 24105,
Germany
*Correspondence to*: Meixun Zhao (maxzhao@ouc.edu.cn)





## Abstract

Climate-driven changes in environmental conditions have significant and complex
effects on marine ecosystems. Variability in phytoplankton elements and biochemicals
can be important for global ocean biogeochemistry and ecological functions, while
there is currently limited understanding on how elemental stoichiometry and
biochemicals respond to the changing environments in key coccolithophore species
such as *Emiliania huxleyi*. We investigated responses of stoichiometric C:N:P ratios,
PIC:POC contents and ratios, and fatty acid (FA) composition in a strain of *E. huxleyi*
under three temperatures (12, 18 and 24 °C), three N:P supply ratios (10:1, 24:1 and
63:1 mol mol$^{-1}$) and two $p$CO$_2$ levels (560 and 2400 μatm). Overall, C:N:P biomass
ratios showed the most pronounced response to N:P supply ratios, with low N:C and
N:P biomass ratios in low N-media, and low P:C and high N:P biomass ratios in low
P-media. PIC:POC ratios and polyunsaturated FA proportions strongly responded to
temperature and $p$CO$_2$, both being lower under high $p$CO$_2$ and higher with warming.
We observed synergistic interactions between warming and nutrient deficiency (and
high $p$CO$_2$) on PIC and POC cellular contents in most cases, indicating the enhanced
effect of warming on *E. huxleyi* calcification and photosynthesis under nutrient
deficiency (and high $p$CO$_2$). Our results suggest differential sensitivity of elements
and FAs to the changes in temperature, nutrient availability and $p$CO$_2$ in *E. huxleyi*,
which is to some extent unique compared with non-calcifying algal classes. Thus,
simultaneous changes of elements and FAs should be considered when predicting
future roles of *E. huxleyi* in the biotic-mediated connection between biogeochemical



cycles, ecological functions and climate change.
**Key words:** Coccolithophores; elements; biochemicals; warming; nutrients; $CO_2$

















## 1 Introduction


Climate change and intensive anthropogenic pressures have pronounced and
diverse effects on marine ecosystems. Physical and chemical properties in marine
ecosystems are changing simultaneously such as the concurrent shifts in temperature,
$CO_2$ and oxygen concentrations, and nutrient availability (Boyd et al., 2015). These
changes have altered trophic interactions in both bottom-up and top-down directions
and thus result in changes in community structure and ecosystem functions (Doney et
al., 2012). Phytoplankton are the base of marine food webs and major drivers of ocean
biogeochemical cycling, and thus quantifying their responses to changing oceanic
conditions is a major challenge in studies of food web structure and ocean
biogeochemistry.
Coccolithophores are a key phytoplankton group in the ocean because of their
production of calcified scales called coccoliths. They are not only important primary
producers, but also play predominant roles in the oceanic calcification via releasing
$CO_2$ to the atmosphere (Rost and Riebesell, 2004). Thus, coccolithophores have a
complex and significant influence on global carbon cycle, by playing an important
role in ocean-atmosphere exchange of $CO_2$ (Rost and Riebesell, 2004). Of all
coccolithophores, *Emiliania huxleyi* is the most widely distributed and the most
abundant species (Winter et al., 2014), with the capacity to form spatially extensive
blooms in mid- to high-latitudes (Raitsos et al., 2006; Tyrrell and Merico, 2004).
Evidence from *in situ* and satellite observations indicates that *E. huxleyi* is
increasingly expanding its range poleward in both hemispheres over the last two



decades, and contributing factors to this poleward expansion may differ between
regions and hemispheres (Winter et al., 2014). For example, nutrients and dissolved
inorganic carbon (DIC) were positively correlated with the increase in
coccolithophore abundance in the subtropical North Atlantic (Krumhardt et al., 2016),
while temperature and irradiance were best able to explain variability in *E.*
*huxleyi*-dominated coccolithophore community composition and abundance across the
Drake Passage (Southern Ocean) (Charalampopoulou et al., 2016). Hence, empirical
data on the responses of *E. huxleyi* to different environmental drivers would be critical
for fully understanding the roles of this prominent coccolithophore species in marine
ecosystems.

Extensive experimental studies have shown highly variable responses of *E. huxleyi*

to rising atmospheric $CO_2$ (reviewed by Feng et al., 2017; Meyer and Riebesell, 2015),
while other studies focused on the influence of other environmental factors such as
temperature (Rosas-Navarro et al., 2016; Sett et al., 2014; Sorrosa et al., 2005), light
intensity (Nanninga and Tyrrell, 1996; Xing et al., 2015) and nutrient availability
(Oviedo et al., 2014; Paasche, 1998). Responses of *E. huxleyi* to the interactions
between these different factors have recently received more attention (De Bodt et al.,
2010; Feng et al., 2008; Milner et al., 2016; Perrin et al., 2016; Rokitta and Rost,
2012). Many of these studies above focused on the physiological, calcification and
photosynthetic responses of *E. huxleyi* due to its considerable role in global carbon
cycle. However, biogeochemical cycles of the major nutrient elements (nitrogen and
phosphorus) and carbon are tightly linked (Hutchins et al., 2009), and thus variability

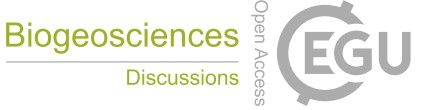

in stoichiometric C:N:P ratios (cellular quotas and ratios of C, N and P) in *E. huxleyi*
can also be a core feature of ocean biogeochemistry. Moreover, element budgets in
organisms are primarily determined by the physiology and biochemistry of
biochemicals such as proteins and fatty acids (FAs) (Anderson et al., 2004; Sterner
and Elser, 2002). Thus, studying simultaneous changes of elements and biochemicals
enables the connection between climate change and ecosystem functions such as
elemental cycles; however, the role of nutrient content of food is often overlooked in
climate change ecology (Rosenblatt and Schmitz, 2016). Recently, Bi et al. (2017)
investigated responses of stoichiometric C:N:P and FAs to the interactions of three
environmental factors in the diatom *Phaeodactylum tricornutum* and the cryptophyte
*Rhodomonas* sp., showing dramatic effects of warming and nutrient deficiency, and
modest effects of increased $p$CO$_2$. However, for the key coccolithophore species *E.*
*huxleyi* much less is known about the simultaneous changes in stoichiometric C:N:P
ratios, calcification, photosynthesis and FAs in response to multiple environmental
factor changes.

In the present study, we conducted semi-continuous cultures of *E. huxleyi* to

disentangle potential effects of temperature, N:P supply ratios and $p$CO$_2$ on *E. huxleyi*
stoichiometric C:N:P ratios, particulate inorganic carbon (PIC) and particulate organic
carbon (POC) contents and their ratios, and FA composition. As the physiological (i.e.,
cellular) PIC and POC variations cannot directly be up scaled to total population
response (Matthiessen et al., 2012), responses of PIC and POC contents in our study
were shown both on the cellular (as pg cell$^{-1}$) and the population (as μg ml$^{-1}$) levels.



FA data were expressed as a percentage of total fatty acids (TFAs) (FA proportion, %
of TFAs) to better compare our results with those in previous studies. FAs were also
quantified on a per unit biomass ($\mu$g mg $C^{-1}$), which is an ideal approach when
considering nutritional quality of phytoplankton for herbivores (Piepho et al., 2012).
Specifically, we addressed the following two questions in the present study, (i) what is
the sensitivity of stoichiometric C:N:P ratios, PIC:POC contents and their ratios, and
FAs of *E. huxleyi* to changes in temperature, N:P supply ratios and $pCO_2$? (ii) how do
stoichiometric C:N:P ratios, PIC:POC and FAs of *E. huxleyi* respond to the
interactions between the three environmental factors?
**2 Material and methods**
**2.1 Experimental setup**

To address our questions on how multiple environmental drivers influence

elemental and FA composition in *E. huxleyi*, we performed a semi-continuous culture
experiment crossing three temperatures (12, 18 and 24 ℃), three N:P supply ratios
(10:1, 24:1 and 63:1 mol $mol^{-1}$) and two $pCO_2$ levels (560 and 2400 $\mu$atm) with a
strain of *E. huxleyi* originating from waters off Terceira Island (Azores, North
Atlantic). The target values of temperature, N:P supply ratios and $pCO_2$ were chosen
to reflect a present natural regime and future ocean projections of each factor
(Boersma et al., 2016; Lewandowska et al., 2014; Moore et al., 2013; IPCC 2014;
Pe ñuelas et al., 2012).

All cultures were exposed to a light intensity of 100 $\mu$mol photons · $m^{-2}$ · $s^{-1}$ at a

16:8 h light:dark cycle in temperature-controlled rooms. The culture medium was





prepared with sterile filtered (0.2 μm pore size, Sartobran® P 300; Sartorius,
Goettingen, Germany) North Sea water with a salinity of 37 psu. Macronutrients were
added as sodium nitrate ($NaNO_3$) and potassium dihydrogen phosphate ($KH_2PO_4$) to
achieve three N:P supply ratios, i.e., 35.2 μmol $\cdot L^{-1}$ N and 3.6 μmol $\cdot L^{-1}$ P (10:1 mol
$mol^{-1}$), 88 μmol $\cdot L^{-1}$ N and 3.6 μmol $\cdot L^{-1}$ P (24:1 mol $mol^{-1}$) and 88 μmol $\cdot L^{-1}$ N and
1.4 μmol $\cdot L^{-1}$ P (63:1 mol $mol^{-1}$). Vitamins and trace metals were added based on the
modified Provasoli's culture medium (Ismar et al., 2008; Provasoli, 1963). Initial
$p$CO$_2$ of the culture medium was manipulated by bubbling with air containing the
target $p$CO$_2$. Three replicates were set up for each treatment, resulting in 54
experimental units. Each culture was kept in a sealed cell culture flask with 920 mL
culture volume. Culture flasks were carefully rotated twice per day at a set time to
minimize sedimentation.
First, batch culture experiments were performed to obtain an estimate of the
observed maximal growth rate ($\mu_{max}$, day$^{-1}$) under three temperatures, three N:P
supply ratios and two $p$CO$_2$ levels. $\mu_{max}$ was calculated based on the changes of
population cell density within exponential phase (Bi et al., 2012). Once batch cultures
reached the early stationary phase, semi-continuous cultures were started with the
algae from batch cultures. The specific growth rate of 20% of $\mu_{max}$ ($\mu$, day$^{-1}$) was
applied. The equivalent daily renewal rate ($D$, day$^{-1}$) can be calculated according to
the equation $D = 1- e^{-\mu t}$, where $t$ is renewal interval (day) (here $t$ = 1 day). The
incubation water was exchanged with fresh filtered seawater enriched by
macronutrients and micronutrients according to the target N:P supply ratios, as well as





pre-acclimated to the desired $p$CO$_2$ level. To counterbalance the biological
CO$_2$-drawdown, the required amount of CO$_2$-saturated seawater was also added.
Renewal of the cultures was carried out at the same hour every day. The steady state
in semi-continuous cultures was assessed based on the net growth rate ($r$). When $r$
was zero (at steady state), μ was equivalent to $D$.

**2.2 Sample analysis**

Sampling took place at steady state for the following parameters: cell density, DIC,
total alkalinity (TA), pH, total particulate carbon (TPC), POC, particulate organic
nitrogen and phosphorus (PON and POP), and FAs. Cell density was counted daily in
batch and semi-continuous cultures. pH measurements were conducted daily in
semi-continuous cultures (Fig. S1), and the electrode was calibrated using standard
pH buffers (pH 4 and pH 7; WTW, Weilheim, Germany).
DIC water samples were gently filtered using a single-use syringe filter (0.2 μm,
Minisart RC25; Sartorius, Goettingen, Germany) which was connected to the intake
tube of a peristaltic pump. Samples were collected into 10 ml glass vials, and all vials
were immediately sealed after filling. DIC was analyzed following Hansen et al.
(2013) using a gas chromatographic system (8610C; SRI-Instruments, California,
USA). Samples for TA analysis were filtered through GF/F filters (Whatman GmbH,
Dassel, Germany) and analyzed with the Tirino plus 848 (Metrohm, Filderstadt,
Germany). The remaining carbonate parameter $p$CO$_2$ was calculated using CO2SYS
(Pierrot et al., 2006) and the constants supplied by Hansson (1973) and Mehrbach et
al. (1973) that were refitted by Dickson and Millero (1987) (Table S1).



199 TPC, POC, PON and POP samples were filtered onto pre-combusted and

200 pre-washed (5%~10% HCl) GF/F filters (Whatman GmbH, Dassel, Germany). For

201 POC samples, PIC was removed by exposing filters containing TPC to fuming

202 hydrochloric acid for 12h. Before analysis, filters were dried at 60 °C and stored in a

203 desiccator. POC and PON was simultaneously determined by gas chromatography in

204 an organic elemental analyzer (Thermo Flash 2000; Thermo Fisher Scientific Inc.,

205 Schwerte, Germany) after Sharp (1974). POP was analyzed colorimetrically by

206 converting organic phosphorus compounds to orthophosphate (Hansen and Koroleff,

207 1999). PIC was determined by subtracting POC from TPC. PIC and POC production

208 were estimated by multiplying μ with cellular PIC or POC content, respectively.

209 FA samples were taken on pre-combusted and hydrochloric acid-treated GF/F

210 filters (Whatman GmbH, Dassel, Germany), and stored at -80 °C before measurement.

211 FAs were measured as fatty acid methyl esters (FAMEs) using a gas chromatograph

212 (Trace GC-Ultra; Thermo Fisher Scientific Inc., Schwerte, Germany) according to the

213 procedure described in detail in Arndt and Sommer (2014). The FAME 19:0 was

214 added as internal standard and 21:0 as esterification control. The extracted FAs were

215 dissolved with n-hexane to a final volume of 100 μL. Sample aliquots (1 μL) were

216 given into the GC by splitless injection with hydrogen as the carrier gas. Individual

217 FAs were integrated using Chromcard software (Thermo Fisher Scientific Inc.,

218 Schwerte, Germany) and identified with reference to the standards Supelco 37

219 component FAME mixture and Supelco Menhaden fish oil.




**2.3 Statistical analysis**
Generalized linear mixed models (GLMMs) were applied to test the best model
explaining the variations in elemental and FA composition. In our study, response
variables included stoichiometric C:N:P ratios (as mol mol$^{-1}$), PIC and POC contents
per cell (as pg cell$^{-1}$) and per ml (as μg ml$^{-1}$), PIC and POC production (as pg cell$^{-1}$
d$^{-1}$), PIC:POC ratio, FA proportion (as % of TFAs) and contents (as μg mg C$^{-1}$), with
temperature, N:P supply ratios and $p$CO$_2$ as fixed effects. Target distributions were
tested and link functions were consequently chosen. For all response variables, we
tested models containing first order effects, and second and third order interactions of
the three factors. The model that best predicted targets was selected based on the
Akaike Information Criterion corrected (AICc), i.e., a lower AICc value representing
a better fit of the model. Changes of 10 units or more in AICc values were considered
as a reasonable improvement in the fitting of GLMMs (Bolker et al., 2009). In case
AICc values were comparable (<10 units difference), the simpler model was thus
chosen, unless there were significant second or third order interactions detected.
Differences in AICc values between different models were more than 10 for most
variables, with the exception for N:P biomass ratio, cellular PIC and POC contents,
and saturated fatty acid (SFA) and DHA proportions, which were less than 10 between
different models (Table S2).
Two factorial ANOVA was used to test the effects of temperature and $p$CO$_2$ on μ$_{max}$.
As μ$_{max}$ should be equal across different nutrients (Cherif and Loreau, 2010), the
effect of N:P supply ratios on μ$_{max}$ was not tested. The normality of dependent



243 variables was checked using the Shapiro-Wilk's *W*-test. For the significant factors, the

244 magnitude of effect ($\omega^2$ = (effect sum of squares – effect degree of freedom × error

245 mean square) / (total sum of squares + error mean square)) was calculated to

246 determine the variance in a response variable and to relate this to the total variance in

247 the response variable (Graham and Edwards, 2001; Hughes and Stachowicz, 2009).

248  Nested models were applied to test whether the response pattern to one factor (a

249 nested factor) was significant within another factor, in case significant second order

250 interactions were detected in GLMM. Also, the nature (antagonistic, additive, or

251 synergistic) of significant second order interactions was analysed according to

252 Christensen et al. (2006). The observed combined effect of two factors was compared

253 with their expected net additive effect [e.g., ($factor_1$ - control) + ($factor_2$ - control)],

254 which was based on the sum of their individual effects. If the observed combined

255 effect exceeded their expected additive effect, the interaction was defined as

256 synergism. In contrast, if the observed combined effect was less than the additive

257 effect, the interaction was defined as antagonism.

258  All statistical analyses were conducted using SPSS 19.0 (IBM Corporation, New

259 York, USA). Significance level was set to $p < 0.05$ in all statistical tests.








## 3 Results

### 3.1 Maximal growth rate ($\mu_{max}$)

The observed maximal growth rate of *E. huxleyi* responded significantly to temperature and the interactions between temperature and $p$CO$_2$ (Bold letters in Table 1), with temperature causing 82% of the effect magnitude. Increasing temperature stimulated $\mu_{max}$ in each $p$CO$_2$ treatment, causing $\mu_{max}$ to be two to three times higher at the highest temperature than those at the lowest temperature (Fig. 1). In contrast, the effect of elevated $p$CO$_2$ on $\mu_{max}$ showed no consistent pattern, indicating a slight positive effect at 12°C, a strong negative effect at 18°C and a weak negative effect at 24°C (Fig. 1). Moreover, the trend of $\mu_{max}$ to increase with increasing temperature differed between the two $p$CO$_2$ treatments, showing that at the low $p$CO$_2$, the slope of $\mu_{max}$ response to increasing temperature was higher from 12°C to 18°C and it became lower from 18°C to 24°C, while the slope of $\mu_{max}$ response showed no clear difference between three temperatures at the high $p$CO$_2$ treatment.

### 3.2 C:N:P stoichiometry

GLMMs results showed that N:C, P:C and N:P biomass ratios responded significantly to N:P supply ratios (Bold letters in Table 2), while only N:C biomass ratios showed significant responses to temperature, with non-significant effect of $p$CO$_2$ detected. Increasing N:P supply ratios caused an increased trend in N:C biomass ratios (Fig. 2a) and a decrease in P:C biomass ratios (Fig. 2b), resulting in a positive relationship between N:P biomass ratios and N:P supply ratios (Fig. 2c). The response of N:C biomass ratios to increasing temperature was complex, showing a U-shaped



response under N deficiency (N:P supply ratio = 10:1 mol mol$^{-1}$), and positive
responses under higher N:P supply ratios (Fig. 2a).

All three PIC-related variables (cellular PIC contents, population yields of PIC and

PIC production) responded significantly to temperature and $p$CO$_2$; and there were also
significant interactions between temperature and $p$CO$_2$ (and N:P supply ratios) on
cellular PIC contents, and significant effects of N:P supply ratios on population yields
of PIC and PIC production (Table 2). Increasing temperature and N deficiency
affected cellular PIC contents antagonistically, while increasing temperature and P
deficiency (N:P supply ratio = 63:1 mol mol$^{-1}$) had synergistic interactions on cellular
PIC contents (Table S3). As a result, cellular PIC contents showed a trend to decrease
slightly with increasing temperature under N deficiency and a trend to increase under
higher N:P supply ratios (Fig. 3a). Increasing temperature and enhanced $p$CO$_2$
affected cellular PIC contents synergistically (Table S3), with the negative response of
cellular PIC contents to enhanced $p$CO$_2$ being significantly weaker as temperature
increased (Fig. 3d; Nested model, $p < 0.001$). Population yields of PIC decreased
(under higher temperatures), but PIC production increased with increasing N:P supply
ratios (Fig. 3b, c), while both PIC variables increased with increasing temperature and
decreased with enhanced $p$CO$_2$ (Fig. 3e, f).

The three POC-related variables (cellular POC contents, population yields of POC

and POC production) responded significantly to temperature; and there were also
significant interactions between temperature and N:P supply ratios on cellular POC
contents, and significant effects of N:P supply ratios on population yields of POC and





POC production (Table 2). For cellular POC contents, increasing temperature and N
(and P) deficiency showed synergistic interactions, resulting in the highest values at
the lowest temperature under lower N:P supply ratios and an increasing trend with
increasing temperature under P deficiency (Fig. 4a; Table S3). Population yields of
POC showed an unimodal response and POC production showed a trend to increase in
response to increasing temperature and N:P supply ratios (Fig. 4b, c).
PIC:POC responded significantly to temperature and $p$CO$_2$ (Table 2), showing a
clear increase with increasing temperature and a decrease with enhanced $p$CO$_2$ (Fig.

5).

**3.3 Fatty acids**

The most abundant FA group were polyunsaturated fatty acids (PUFAs) (33%-54%
of TFAs), followed by SFAs (22%-46%) and monounsaturated fatty acids (MUFAs)
(13%-27%), across the entire tested gradients of temperature, N:P supply ratios and
$p$CO$_2$ (Table S4). The high proportion of PUFAs was predominantly caused by high
amounts of DHA (12%-31%) and 18:4n-3 (3%-13%), and SFAs was mainly
represented by 14:0 (13%-23%) and 16:0 (5%-11%). The major individual MUFA
was 18:1n-9 (8%-21%).
GLMMs results showed significant effects of temperature and $p$CO$_2$ on the
proportions of both MUFAs and PUFAs, and significant interactions between N:P
supply ratios and $p$CO$_2$ on SFAs (Table 2). Increasing temperature caused a decrease
in the proportion of MUFAs and an increase in PUFAs (Fig. 6a, b). In contrast,
enhanced $p$CO$_2$ resulted in an increase in MUFAs and a decrease in PUFAs at higher



temperatures (Fig. 6d, e). Moreover, enhanced $pCO_2$ and N (and P) deficiency
affected SFA proportion synergistically (Table S3), with the unimodal response of
SFA proportion to increasing N:P supply ratios being more pronounced at the high
$pCO_2$ (Fig. S2; Nested model, $p < 0.001$).
The proportion of major individual PUFAs (DHA) showed significant responses to
temperature and N:P supply ratios, and the interactions between temperature and N:P
supply ratios (and $pCO_2$) (Table 2). Increasing temperature caused an overall increase
in DHA, and DHA also had higher values under N and P deficiency (Fig. 6c). The
interactions between increasing temperature and N deficiency (and P deficiency and
enhanced $pCO_2$) affected DHA synergistically (Table S3), and the positive effect of
temperature became more pronounced at lower N:P supply ratios (Nested model, $p <$
$0.001$) and at the low $pCO_2$ (Nested model, $p < 0.001$) (Fig. 6c, f).

**4 Discussion**

Our study scales the impacts of temperature, N:P supply ratios and $pCO_2$ on
elemental and FA composition of the ubiquitously important calcifier *E. huxleyi*, while
accounting for their interactive effects. Overall, C:N:P biomass ratios changed
markedly in response to N:P supply ratios, resulting in up to a 62% increase in N:P
biomass ratios (Fig. 7). PIC:POC showed an increase of 41% with warming and a
decrease of 35% under high $pCO_2$. PUFA proportions showed an increase of 13%
with warming and a decrease of 7% with enhanced $pCO_2$, indicating a partial
compensation by $pCO_2$ of a predominantly temperature-driven response. The overall





response patterns of C:N:P stoichiometry and PUFAs in our study are consistent with
those on the global scale (Martiny et al., 2013), and conform with the meta-analysis
results on haptophytes (Hixson and Arts, 2016). In line with these studies, we also
detected significant interactions between temperature, N:P supply ratios and $p$CO$_2$,
indicating variable response patterns of elemental and FA composition in *E. huxleyi*
under any given constellation of environmental factors. Our results thus underscore
the importance of simultaneous consideration of multiple environmental drivers,
demonstrating differential effects of the three environmental factors on elemental and
FA composition of *E. huxleyi*.
**4.1 Responses of maximal growth rate**

Increasing temperature (12-24°C) significantly accelerated $\mu_{max}$ of *E. huxleyi* in our

study (Fig. 1). This positive correlation between increasing temperature and growth
rate is typical for many *E. huxleyi* strains within the range of temperature 12 to 24°C
(Feng et al., 2008; Rosas-Navarro et al., 2016; Sett et al., 2014; van Bleijswijk et al.,
1994). However, the extent to which growth rate of *E. huxleyi* increases with
increasing temperature varies between *E. huxleyi* strains, which may contribute to
specific biogeographic distribution of different strain (Paasche, 2002). For example,
growth rate of *E. huxleyi* from the Gulf of Maine (~42 °N) was 1.2 times higher at 26°C
than at 16°C, while growth rate of *E. huxleyi* from the Sargasso Sea (~20-35 °N) was
1.6 times higher at the higher temperature (Paasche, 2002). Also, our results showed
that $\mu_{max}$ of *E. huxleyi* was two to three times higher at the highest temperature than at
the lowest temperature, suggesting that temperature is an important environmental





factor in controlling the distribution of this strain of *E. huxleyi*.
Also, temperature showed significant interactions with $pCO_2$ on $\mu_{max}$ of *E. huxleyi*
in this study. On the one hand, elevated $pCO_2$ led to an increase in $\mu_{max}$ at the lowest
temperature (12℃) but a decrease at higher temperatures (18 and 24℃) (Fig. 1).
Similarly, in response to increasing $pCO_2$, either an increase (Feng et al., 2008) or a
decrease (Milner et al., 2016) in the growth of *E. huxleyi* was observed in previous
studies. Such a diverse response of phytoplankton growth rate to elevated $pCO_2$ has
been widely observed within functional groups such as coccolithophores and diatoms,
between taxa and even between strains of the same species (Dutkiewicz et al., 2015).
Significant interactions between $pCO_2$ and temperature on *E. huxleyi* growth in our
study suggest that experimental temperature can be an important factor resulting in
diverse responses of phytoplankton growth to rising $pCO_2$ in previous studies.
On the other hand, we observed different slopes of $\mu_{max}$ in response to increasing
temperature at two $pCO_2$ levels, showing a decrease in the slope with increasing
temperature at the low $pCO_2$ and a relatively constant slope at the high $pCO_2$ (Fig. 1).
Our results are consistent with a conceptual graph proposed by Sett et al. (2014). The
graph showed a clear increase in metabolic rates from low to intermediate temperature
and a slight increase from intermediate to high temperature at the low $pCO_2$ (~560
µatm), while and the changes of metabolic rates are similar from low to intermediate
temperature and from intermediate to high temperature at the high $pCO_2$ (~2400 µatm)
(Sett et al., 2014). While the physiological mechanisms governing adaptation of
phytoplankton to rising $pCO_2$ are still unclear, one possible explanation is that





increasing temperature may modulate the balance between a fertilizing effect of ocean
carbonation and a metabolic repression by ocean acidification (Bach et al., 2011; Sett
et al., 2014).

**4.2 Responses of C:N:P biomass ratios**

**4.2.1 Effects of N:P supply ratios**

N:P supply ratios showed highly significant effects on N:C, P:C and N:P biomass
ratios in *E. huxleyi* in this study, contributing up to 62% changes in C:N:P biomass
ratios (Table 2; Fig. 7). The strong contribution of nutrient availability to the
elemental ratios of marine phytoplankton community biomass was also found on the
global scale (Daines et al., 2014; Martiny et al., 2013), with nitrate concentration as a
proxy of nutrient availability explaining 36% of variation in N:P ratio and 42% of
variation in C:P ratio (Martiny et al., 2013). Similarly, previous lab experiments
reported that nutrient availability prevailed the governing effect on stoichiometric
ratios in *E. huxleyi* (Skau, 2015).
Overall, N deficiency caused low N:C and N:P biomass ratios, while P deficiency
resulted in low P:C biomass ratios and high N:P biomass ratios in *E. huxleyi* in this
and most previous studies (Langer et al., 2013; Leonardos and Geider, 2005b; Perrin
et al., 2016). An important biogeochemical question is the extent to which N:P
biomass ratios change under N and P deficiency, respectively. Our results showed that
changes in N:P biomass ratios of *E. huxleyi* were mostly due to P deficiency (a 62%
increase), with smaller changes (a 36% decrease) induced by N deficiency (Fig. 7).
This observation is consistent with the high variability of P:C in response to changes



in phosphate and the less variable N:C to changes in nitrate based on global
suspended particle measurements (Galbraith and Martiny, 2015). Indeed, conflicting
response patterns of C:N:P biomass ratios of *E. huxleyi* to nutrient deficiency were
observed between different studies (Borchard and Engel, 2012; Matthiessen et al.,
2012; Perrin et al., 2016), which could be due to strain-specific responses and
interactions between nutrients and other environmental drivers. Nevertheless, the
nutrient-dependence of C:N:P biomass ratios of *E. huxleyi* in our study is consistent
with responses of marine plankton on the global scale, which may reflect the capacity
of this species to thrive under a wide range of environmental conditions. This capacity
was largely revealed by a pan-genome assessment, which distributed genetic traits
variably between strains and showed a suit of core genes for the uptake of inorganic
nitrogen and N-rich compounds such as urea (Read et al., 2013). In spite of strain
diversity within *E. huxleyi*, a recent study suggested that the global physiological
response of this species to nutrient environments is highly conserved across strains
and may underpin its success under a variety of marine environments (Alexander,

434    2016).

**4.2.2 Effects of temperature**
Temperature had a weaker effect (5-8% changes) than N:P supply ratios on
variation of N:P and P:C ratios (Table 2; Fig. 7), consistent with results in marine
plankton communities on the global scale (Martiny et al., 2013). While both N:C and
P:C biomass ratios increased with increasing temperature in our study, the changes in
N:C ratios (8%) were larger than those in P:C ratios (5%). As a result, the observed



N:P ratios also increased in response to warming, in accordance with the positive
relationship between temperature and N:P ratios of marine phytoplankton sampled
from different biogeochemical provinces (Martiny et al., 2016; Toseland et al., 2013;
Yvon-Durocher et al., 2015). These responses are consistent with proposed
physiological mechanisms (Toseland et al., 2013), which showed that eukaryotic
phytoplankton at higher temperatures required less P-rich ribosomes and thus
produced higher N:P ratios.
**4.2.3 Effects of $p$CO$_2$**
Partial CO$_2$ pressure showed a non-significant effect on *E. huxleyi* C:N:P biomass
ratios in our study (Table 2), being consistent with the previous findings of the less
important effect of $p$CO$_2$ than nutrients and temperature on *E. huxleyi* (Boyd et al.,
2010; Feng, 2015). Several experimental studies also showed non-significant changes
in *E. huxleyi* C:N:P stoichiometry in response to rising $p$CO$_2$ (Engel et al., 2014;
Lefebvre et al., 2012; Olson et al., 2017); however, some studies reported clear
changes in C:N or C:P ratio (Engel et al., 2005; Leonardos and Geider, 2005a;
Matthiessen et al., 2012). Both experimental and model studies have suggested that
rising $p$CO$_2$ seems to change phytoplankton stoichiometry under specific conditions,
e.g., at high light condition (Feng et al., 2008) and at low nutrient loads (Leonardos
and Geider, 2005a; Verspagen et al., 2014). Moreover, the sensitivity of phytoplankton
to CO$_2$ may also depend on culture systems, either transient or sufficiently stable,
influencing optimal allocation of energy and resources (Engel et al., 2014).
Taken together, our results indicate that N:P supply ratios are reflected in elemental



make-up of cell biomass in *E. huxleyi*, across different temperatures and $pCO_2$ levels,
showing the absence of significant interactions between the three environmental
factors. However, for two algal species from non-calcifying classes (the diatom *P.*
*tricornutum* and the cryptophyte *Rhodomonas* sp.) temperature had the most
consistent significant effect on N:C and P:C biomass ratios and showed significant
interactions with N:P supply ratios and $pCO_2$ in our previous work (Bi et al., 2017).
Both temperature and N:P supply ratios were also ranked as important factors in
regulating phytoplankton stoichiometry in previous studies (Boyd et al., 2010; Feng,
2015). Differential C:N:P responses to environmental drivers between phytoplankton
groups suggest that taxonomic composition may explain the variable C:N:P ratios in
surface phytoplankton community (Martiny et al., 2013).
**4.3 Responses of PIC:POC contents and ratios**
Both partial $CO_2$ pressure and temperature had highly significant effects on
PIC:POC in our study, with enhanced $pCO_2$ resulting in an overall 49% decrease in
PIC:POC and warming resulting in a 41% increase in PIC:POC, while N:P supply
ratios showed no significant effect (Table 2; Fig. 7). This result is in agreement with
rankings of the importance of environmental drivers on PIC:POC in a Southern
Hemisphere strain of *E. huxleyi*, showing the order of $pCO_2$ (negative effect) >
temperature (positive effect) and non-significant effect of nitrate or phosphate (Feng,

482    2015).

The negative effect of enhanced $pCO_2$ on PIC:POC was also observed for different
strains of *E. huxleyi* in most previous studies (Meyer and Riebesell, 2015 and



references therein). Negative responses of PIC:POC to increasing $p$CO$_2$ in our study
were driven by the significant decrease in the three PIC-related response variables
(calcification) and no significant change in all three variables of POC (photosynthesis)
(Table 2; Fig. 7). To date, studies and reviews also showed a greater impact of ocean
acidification on calcification than on photosynthesis in coccolithophores (De Bodt et
al., 2010; Feng et al., 2017; Meyer and Riebesell, 2015). Feng et al. (2017) suggested
that the decreased calcification in *E. huxleyi* may be caused by the increased
requirement of energy to counteract intracellular acidification. And the increased
activity of carbonic anhydrase (CA) at low $p$CO$_2$ may explain the lack of a significant
effect of $p$CO$_2$ on the photosynthetic or growth rate (Feng et al., 2017), as
up-regulation of CA at low DIC was previously observed (Bach et al., 2013).
A positive response or no clear change in PIC:POC was also observed for *E.*
*huxleyi* in response to high $p$CO$_2$ in previous studies (Feng et al., 2017), with the
influence of other environmental drivers proposed as a potential driver (De Bodt et al.,
2010; Feng et al., 2017; Feng et al., 2008). Indeed, we observed that the negative
relationship between cellular PIC contents and enhanced $p$CO$_2$ became weaker at the
highest temperature (Fig. 3d). This result is in agreement with the modulating effect
of temperature on the CO$_2$ sensitivity of key metabolic rates in coccolithophores, due
to the shift of the optimum CO$_2$ concentration for key metabolic processes towards
higher CO$_2$ concentrations from intermediate to high temperatures (Sett et al., 2014).
Temperature causes diverse responses of calcification and photosynthesis within *E.*
*huxleyi* species in the literature (Rosas-Navarro et al., 2016 and references therein)

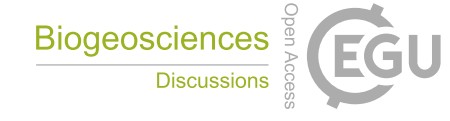

and the present study. Overall, our study showed that the positive response of
PIC:POC to increasing temperature was driven by a marked increased PIC
(28%~161%), and by a less pronounced change in POC (-8%~68%) (Table 2; Fig. 7).
The overall responses of PIC:POC contents and ratios to increasing temperature was
consistent with those in other *E. huxleyi* strains (Matson et al., 2016; Sett et al., 2014),
indicating carbon allocation to calcification rather than photosynthesis at high
temperatures (Sett et al., 2014). Specifically, we observed that the interactions of
warming and nutrient deficiency (and high $p$CO$_2$) synergistically affected both PIC
and POC cellular contents in most cases, suggesting that nutrient deficiency and high
$p$CO$_2$ are likely to enhance the effect of warming on *E. huxleyi* calcification and
photosynthesis efficiency.
In summary, our results showed an overall reduced PIC:POC in *E. huxleyi* under
future ocean scenarios of warming and higher CO$_2$ (Fig. 5b), consistent with the
reduced ratio of calcium carbon production to organic carbon during the *E. huxleyi*
bloom in previous mesocosm experiments (Delille et al., 2005; Engel et al., 2005). It
is worth noting that cellular PIC and POC contents are a measure for physiological
response and cannot be used to infer population response and thus potential carbon
export (Matthiessen et al., 2012), as different responses between cellular and
population yields of PIC (and POC) to environmental changes were evident in
previous work (Matthiessen et al., 2012) and the present study (Fig. 7). Moreover,
significant interactions between temperature and $p$CO$_2$ (and N:P supply ratios) on
cellular particulate carbon contents suggest that the effects of multiple environmental



drivers should be tested to better generalize the dynamic of particulate carbon from
laboratory observations to that in natural conditions.
**4.4 Responses of fatty acids**
**4.4.1 Effects of temperature**
Temperature had highly significant effects on the proportions of MUFAs, PUFAs
and DHA, with increasing temperature causing a 20% decline in MUFAs and a 13%
increase in PUFAs in our study (Table 2; Fig. 7). This result is consistent with the
negative response of MUFA proportions and positive response of PUFAs to warming
in other haptophytes based on a meta-analysis on 137 FA profiles (Hixson and Arts,
2016), showing an opposite response to general patterns of phytoplankton FAs to
warming. Although warming is expected to have a negative effect on the degree of
fatty acid unsaturation to maintain cell membrane structural functions (Fuschino et al.,
2011; Guschina and Harwood, 2006; Sinensky, 1974), variable FA responses to
warming were widely observed in different phytoplankton groups (Bi et al., 2017;
Renaud et al., 2002; Thompson et al., 1992). Contradictory findings were even
reported in meta-analyses on large FA profiles such as the absence (Galloway and
Winder, 2015) or presence (Hixson and Arts, 2016) of the negative correlation
between temperature and the proportion of long-chain EFAs in freshwater and marine
phytoplankton. While the underling mechanisms of variable FA responses are still
unclear, it is known that both phylogeny and environmental conditions determine
phytoplankton FA composition (Bi et al., 2014; Dalsgaard et al., 2003; Galloway and
Winder, 2015). In our study, we found significant interactions between temperature





and $p$CO$_2$ (and N:P supply ratios) on the individual FA component DHA, showing that
$p$CO$_2$ and nutrient availability may alter the effect of warming on *E. huxleyi* FA
composition.
**4.4.2 Effects of $p$CO$_2$**
Partial CO$_2$ pressure had significant effects on the proportion of MUFAs and
PUFAs in *E. huxleyi* in our study, with enhanced $p$CO$_2$ causing an overall 7% increase
in MUFAs and a 7% decrease in PUFAs (Table 2; Fig. 7), consistent with FA response
patterns in another strain of *E. huxleyi* (Riebesell et al., 2000) and other
coccolithophores (Fiorini et al., 2010). Also in a natural plankton community
(Raunefjord, southern Norway), PUFA proportion was reduced at high $p$CO$_2$ level in
the nano-size fraction, suggesting a reduced Haptophyta (dominated by *E. huxleyi*)
biomass and a negative effect of high $p$CO$_2$ on PUFA proportion (Bermúdez et al.,
2016). To date, several mechanisms have been suggested to explain the reduced
PUFAs at high $p$CO$_2$ in green algae (Pronina et al., 1998; Sato et al., 2003; Thompson,
1996), with much less work conducted in other phytoplankton groups. One possible
mechanism was demonstrated in the study on *Chlamydomonas reinhardtii*, showing
that the repression of the CO$_2$-concentrating mechanisms (CCMs) was associated with
reduced FA desaturation at high CO$_2$ concentration (Pronina et al., 1998). Similarly in
our study, both the proportion and contents of PUFAs decreased at high $p$CO$_2$ (Fig. 7),
which may be attributed to the repression of CCMs at high $p$CO$_2$ in *E. huxleyi*.




### 4.4.3 Effects of N:P supply ratios


Significant effects of N:P supply ratios was only observed for the proportion of
DHA in this study (Table 2), with N and P deficiency causing a 14% and 22%
increase in DHA proportion, respectively (Fig. 7). While nutrients often play a major
role on phytoplankton lipid composition (Fields et al., 2014; Hu et al., 2008), the less
pronounced effects of nutrient deficiency in our study indicate a unique lipid
biosynthesis in *E. huxleyi*. Indeed, Van Mooy et al. (2009) suggested that *E. huxleyi*
used non-phosphorus betaine lipids as substitutes for phospholipids in response to P
scarcity. Genes are also present in the core genome of *E. huxleyi* for the synthesis of
betaine lipids and unusual lipids used as nutritional/feedstock supplements (Read et
al., 2013). Therefore, the lack of significant nutrient effects on most FA groups in *E.*
*huxleyi* in our study may be caused by the functioning of certain lipid substitutions
under nutrient deficiency.
In summary, this study showed the strongest effects of temperature and $p$CO$_2$, a
weaker effect of N:P supply ratios, and non-significant interactions between the three
environmental factors, on the proportions of unsaturated fatty acids in *E. huxleyi*.
Specifically, we observed that PUFA proportions in *E. huxleyi* responded positively to
warming and negatively to enhanced $p$CO$_2$. It should be noted that using different
units to quantify FA composition may cause contradictory results, e.g., an increase in
PUFA proportion (% of TFAs) but an overall decline in PUFA contents per biomass
($\mu$g mg C$^{-1}$) with increasing temperature in our study (Fig. 7; Table S4, S5). Such
biochemical quality differences can translate to higher trophic levels (Rossoll et al.,





2012) and refer to direct effects of environmental changes on low trophic level
consumers, which can be modified by indirect bottom-up driven impacts through the
primary producers (Garzke et al., 2016; Garzke et al., 2017). Similar to *E. huxleyi*,
PUFA contents in two species of non-calcifying classes (*P. tricornutum* and
*Rhodomonas* sp.) responded negatively to warming and positively to N (and P)
deficiency (Bi et al., 2017). However, differential responses were also observed, e.g.,
a significant negative effect of enhanced $p$CO$_2$ on PUFA contents in *E. huxleyi*, but a
non-significant effect of $p$CO$_2$ on PUFA contents in *P. tricornutum* and *Rhodomonas*
sp. (Bi et al., 2017). This different response between phytoplankton groups is in
agreement with findings in mesocosm studies (Bermúdez et al., 2016; Leu et al.,
2013), suggesting that changes in taxonomic composition can cause different
relationship between PUFAs and $p$CO$_2$ in natural phytoplankton community.
**5 Conclusions**
Our study showed that N:P supply ratios had the strongest effect on C:N:P biomass
ratios, while $p$CO$_2$ and temperature played more influential roles on PIC:POC and
PUFA proportions in *E. huxleyi*. The specific response patterns of elemental ratios and
PUFAs have important implications for understanding biogeochemical and ecological
functioning of *E. huxleyi*. The overall low PIC:POC under future ocean scenarios
(warming and enhanced $p$CO$_2$) indicates that carbon production by the strain *E.*
*huxleyi* in our study acts as a carbon sink. This argument is consistent with the finding
that the calcification of coccolithophores decreases with increasing $p$CO$_2$ in the
present ocean and over the past forty thousand years (Beaufort et al., 2011). It should



be noted that carbon production discussed so far are based on responses of cellular or
population PIC:POC in a single strain of *E. huxleyi*, and thus it cannot be used to infer
potential carbon flux between surface ocean and the atmosphere. From the ecological
point of view, the contradictory response patterns of N:C and P:C biomass ratios (an
overall increase) and PUFA contents (an overall decrease) under future ocean
scenarios in our study make the extraction of a generic response signal of nutritional
food quality difficult. Nonetheless, the observations presented here suggest
differential responses of elements and FAs to rising temperature, enhanced $p$CO$_2$ and
nutrient deficiency in *E. huxleyi*, being to some extent unique compared with algal
species from non-calcifying classes. Thus, the role of multiple environmental drivers
under the biodiversity context should be considered to truly estimate the future
functioning of phytoplankton in the changing marine environments.












**Data availability**: data sets are available upon request by contacting Meixun Zhao
(maxzhao@ouc.edu.cn and maxzhao04@yahoo.com).
**Author contribution**: R. Bi, S. Ismar, U. Sommer and M. Zhao designed the
experiments and R. Bi carried them out. R. Bi prepared the manuscript with
contributions from all co-authors.
**Competing interests**: the authors declare that they have no conflict of interest.

**Acknowledgements** The authors thank Thomas Hansen, Cordula Meyer, Bente
Gardeler and Petra Schulz for technical assistance. Birte Matthiessen is gratefully
acknowledged for providing the *E. huxleyi* strain. We thank Dorthe Ozod-Seradj,
Carolin Paul, Si Li, Xupeng Chi and Yong Zhang for their assistance during the
experiments, and Philipp Neitzschel, Kastriot Qelaj and Jens Wernhöner for helping
with DIC analysis. Jessica Garzke is acknowledged for her comments on the
calculation of interaction magnitude. This study was funded by the National Natural
Science Foundation of China (Grant No. 41521064; No. 41506086; No. 41630966),
the Scientific Research Foundation for the Returned Overseas Chinese Scholars, State
Education Ministry (Grant No. [2015]1098), the "111" Project (B13030) and
GEOMAR Helmholtz-Zentrum für Ozeanforschung Kiel. This is MCTL contribution

657   150.






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





Table 1. Results of ANOVA testing for the responses of the observed maximum
growth rate to temperature and $p$CO$_2$ in *Emiliania huxleyi*. Significant $p$ values are
shown in bold.

| Factor | df | $F$ | $p$ | $\omega^2$ |
|---|---|---|---|---|
| Temperature | 2 | 89.842 | **<0.001** | 0.82 |
| $p$CO$_2$ | 1 | 3.808 | 0.075 | |
| Temperature $\times$ $p$CO$_2$ | 2 | 10.638 | **0.002** | 0.09 |





















Table 2. Results of the selected GLMMs testing for the effects of temperature, N:P
supply ratios and $p\mathrm{CO}_2$ on stoichiometric C:N:P biomass ratios, PIC and POC
contents and their ratios, PIC and POC production, and fatty acid proportions in
*Emiliania huxleyi*. Significant $p$ values are shown in bold; T: temperature; N:P: N:P
supply ratio; TFA: total fatty acid; SFA: saturated fatty acid; MUFA: monounsaturated
fatty acid; PUFA: polyunsaturated fatty acid; DHA: docosahexaenoic acid (22:6n-3).

| Variable | Factor | Coefficienct $\pm$SE | $t$ | $p$ |
|---|---|---|---|---|
| N:C biomass ratio (mol mol$^{-1}$) | Intercept | 0.062 $\pm$0.007 | 8.659 | <0.001 |
| | T | 0.001 $\pm$<0.001 | 2.641 | **0.011** |
| | $p\mathrm{CO}_2$ | <0.001 $\pm$<0.001 | -0.240 | 0.811 |
| | N:P | <0.001 $\pm$<0.001 | 4.717 | **<0.001** |
| P:C biomass ratio (mmol mol$^{-1}$) | Intercept | 4.481 $\pm$0.451 | 9.946 | <0.001 |
| | T | 0.016 $\pm$0.021 | 0.773 | 0.443 |
| | $p\mathrm{CO}_2$ | <0.001 $\pm$<0.001 | 0.628 | 0.553 |
| | N:P | -0.037 $\pm$0.005 | -8.154 | **<0.001** |
| N:P biomass ratio (mol mol$^{-1}$) | Intercept | 2.702 $\pm$0.145 | 18.590 | <0.001 |
| | T | 0.001 $\pm$0.007 | 0.157 | 0.876 |
| | $p\mathrm{CO}_2$ | <0.001 $\pm$<0.001 | -0.169 | 0.866 |
| | N:P | 0.016 $\pm$0.001 | 11.200 | **<0.001** |
| PIC (pg cell$^{-1}$) | Intercept | 3.293 $\pm$0.406 | 8.122 | <0.001 |
| | T | -0.067 $\pm$0.021 | -3.193 | **0.003** |
| | $p\mathrm{CO}_2$ | -0.001 $\pm$<0.001 | -5.519 | **<0.001** |
| | N:P | -0.003 $\pm$0.009 | -0.292 | 0.772 |
| | T $\times$ $p\mathrm{CO}_2$ | <0.001 $\pm$<0.001 | 4.584 | **<0.001** |
| | T $\times$ N:P | 0.001 $\pm$<0.001 | 2.340 | **0.024** |
| | $p\mathrm{CO}_2$ $\times$ N:P | <0.001 $\pm$<0.001 | 0.111 | 0.912 |
| PIC (µg ml$^{-1}$) | Intercept | 6.922 $\pm$0.968 | 7.149 | <0.001 |
| | T | 0.201 $\pm$0.045 | 4.442 | **<0.001** |
| | $p\mathrm{CO}_2$ | -0.002 $\pm$<0.001 | -8.955 | **<0.001** |
| | N:P | -0.034 $\pm$0.010 | -3.404 | **0.001** |
| PIC production (pg cell$^{-1}$ d$^{-1}$) | Intercept | -0.689 $\pm$0.105 | -6.581 | <0.001 |
| | T | 0.047 $\pm$0.005 | 9.589 | **<0.001** |
| | $p\mathrm{CO}_2$ | <0.001 $\pm$<0.001 | -5.294 | **<0.001** |
| | N:P | 0.007 $\pm$0.001 | 6.339 | **<0.001** |
| POC (pg cell$^{-1}$) | Intercept | 3.683 $\pm$0.377 | 9.779 | <0.001 |
| | T | -0.089 $\pm$0.020 | -4.577 | **<0.001** |
| | $p\mathrm{CO}_2$ | <0.001 $\pm$<0.001 | -0.929 | 0.358 |
| | N:P | -0.008 $\pm$0.008 | -0.996 | 0.324 |



|  |  |  |  |  |
|---|---|---|---|---|
|  | T $\times$ $p$CO$_2$ | <0.001 $\pm$ <0.001 | 1.886 | 0.066 |
|  | T $\times$ N:P | 0.001 $\pm$ <0.001 | 3.477 | **0.001** |
|  | $p$CO$_2$ $\times$ N:P | <0.001 $\pm$ <0.001 | -0.359 | 0.721 |
| POC ($\mu$g ml$^{-1}$) | Intercept | 13.456 $\pm$ 1.007 | 13.360 | <0.001 |
|  | T | -0.096 $\pm$ 0.047 | -2.045 | **0.046** |
|  | $p$CO$_2$ | <0.001 $\pm$ <0.001 | -0.361 | 0.719 |
|  | N:P | -0.035 $\pm$ 0.010 | -3.436 | **0.001** |
| POC production (pg cell$^{-1}$ d$^{-1}$) | Intercept | -0.261 $\pm$ 0.101 | -2.587 | 0.013 |
|  | T | 0.023 $\pm$ 0.005 | 4.895 | **<0.001** |
|  | $p$CO$_2$ | <0.001 $\pm$ <0.001 | 1.631 | 0.109 |
|  | N:P | 0.007 $\pm$ 0.001 | 6.899 | **<0.001** |
| PIC:POC | Intercept | 0.460 $\pm$ 0.066 | 7.010 | <0.001 |
|  | T | 0.025 $\pm$ 0.003 | 8.184 | **<0.001** |
|  | $p$CO$_2$ | <0.001 $\pm$ <0.001 | -12.837 | **<0.001** |
|  | N:P | <0.001 $\pm$ 0.001 | -0.166 | 0.869 |
| SFA proportion (% of TFAs) | Intercept | 3.506 $\pm$ 0.145 | 24.178 | <0.001 |
|  | T | -0.012 $\pm$ 0.008 | -1.538 | 0.131 |
|  | $p$CO$_2$ | <0.001 $\pm$ <0.001 | -0.238 | 0.813 |
|  | N:P | -0.004 $\pm$ 0.003 | -1.248 | 0.218 |
|  | T $\times$ $p$CO$_2$ | <0.001 $\pm$ <0.001 | 1.816 | 0.076 |
|  | T $\times$ N:P | <0.001 $\pm$ <0.001 | 1.657 | 0.104 |
|  | $p$CO$_2$ $\times$ N:P | <0.001 $\pm$ <0.001 | -2.487 | **0.016** |
| MUFA proportion (% of TFAs) | Intercept | 30.259 $\pm$ 1.344 | 22.518 | <0.001 |
|  | T | -0.579 $\pm$ 0.063 | -9.240 | **<0.001** |
|  | $p$CO$_2$ | 0.001 $\pm$ <0.001 | 2.269 | **0.028** |
|  | N:P | -0.014 $\pm$ 0.014 | -1.050 | 0.299 |
| PUFA proportion (% of TFAs) | Intercept | 32.264 $\pm$ 2.300 | 14.028 | <0.001 |
|  | T | 0.638 $\pm$ 0.107 | 5.949 | **<0.001** |
|  | $p$CO$_2$ | -0.002 $\pm$ 0.001 | -2.769 | **0.008** |
|  | N:P | 0.034 $\pm$ 0.023 | 1.453 | 0.152 |
| DHA proportion (% of TFAs) | Intercept | 2.204 $\pm$ 0.185 | 11.887 | <0.001 |
|  | T | 0.054 $\pm$ 0.010 | 5.611 | **<0.001** |
|  | $p$CO$_2$ | <0.001 $\pm$ <0.001 | 1.874 | 0.067 |
|  | N:P | 0.010 $\pm$ 0.004 | 2.735 | **0.009** |
|  | T $\times$ $p$CO$_2$ | <0.001 $\pm$ <0.001 | -2.946 | **0.005** |
|  | T $\times$ N:P | -0.001 $\pm$ <0.001 | -2.898 | **0.006** |
|  | $p$CO$_2$ $\times$ N:P | <0.001 $\pm$ <0.001 | 1.249 | 0.218 |








**Fig. 1** Responses of the observed maximal growth rate ($\mu_{max}$; mean $\pm$ SE) to temperature and $p$CO$_2$ in *Emiliania huxleyi*.

**Fig. 2** Responses of (a) N:C, (b) P:C and (c) N:P biomass ratios (mean $\pm$ SE) to temperature and N:P supply ratios in *Emiliania huxleyi*.

**Fig. 3** Responses of PIC content (a, d) per cell and (b, e) per ml and (c, f) PIC production (mean $\pm$ SE) to temperature, N:P supply ratios and $p$CO$_2$ in *Emiliania huxleyi*.

**Fig. 4** Responses of POC content (a, d) per cell and (b, e) per ml and (c, f) POC production (mean $\pm$ SE) to temperature, N:P supply ratios and $p$CO$_2$ in *Emiliania huxleyi*.

**Fig. 5** Responses of PIC:POC ratio (mean $\pm$ SE) to temperature, N:P supply ratios and $p$CO$_2$ in *Emiliania huxleyi*.

**Fig. 6** Responses of the proportions of (a, d) monounsaturated fatty acids (MUFAs), (b, e) polyunsaturated fatty acids (PUFAs) and (c, f) docosahexaenoic acid (DHA) (mean $\pm$ SE) to temperature, N:P supply ratios and $p$CO$_2$ in *Emiliania huxleyi*.

**Fig. 7** The changes in C:N:P stoichiometry, PIC and POC contents and their ratios, PIC and POC production, and the proportions and contents of major fatty acid groups and DHA in response to warming, N and P deficiency and enhanced $p$CO$_2$ in *Emiliania huxleyi*. Here, not only significant effects are depicted, but also non-significant and substantial effects on response variables. Significant interactions are presented based on GLMM results in Table 2. Red and blue arrows indicate a mean percent increase and decrease in a given response, respectively.





**Fig. 1**

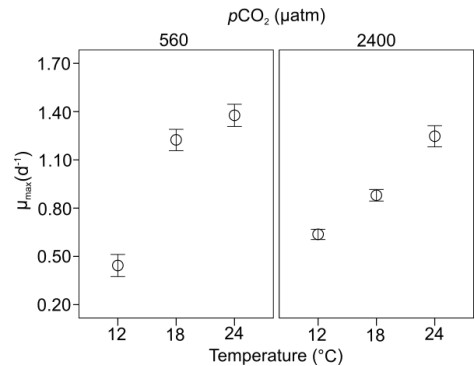




















**Fig. 2**

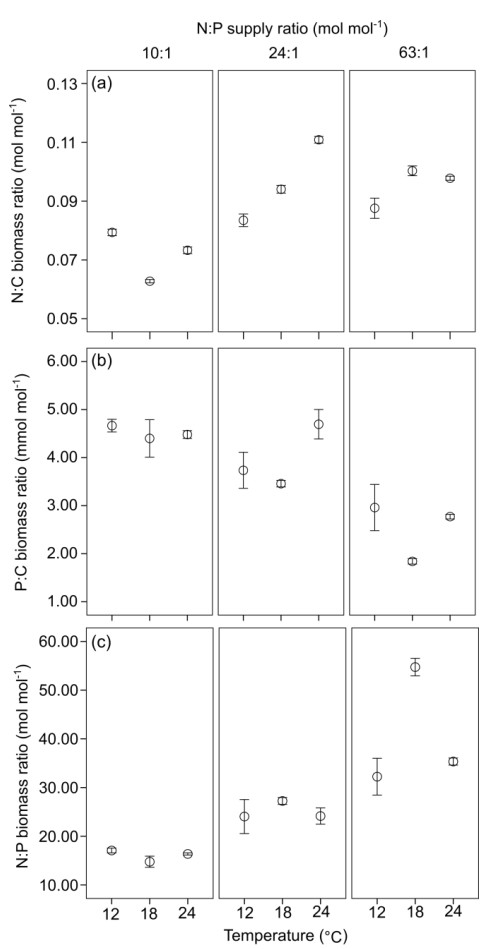











**Fig. 3**

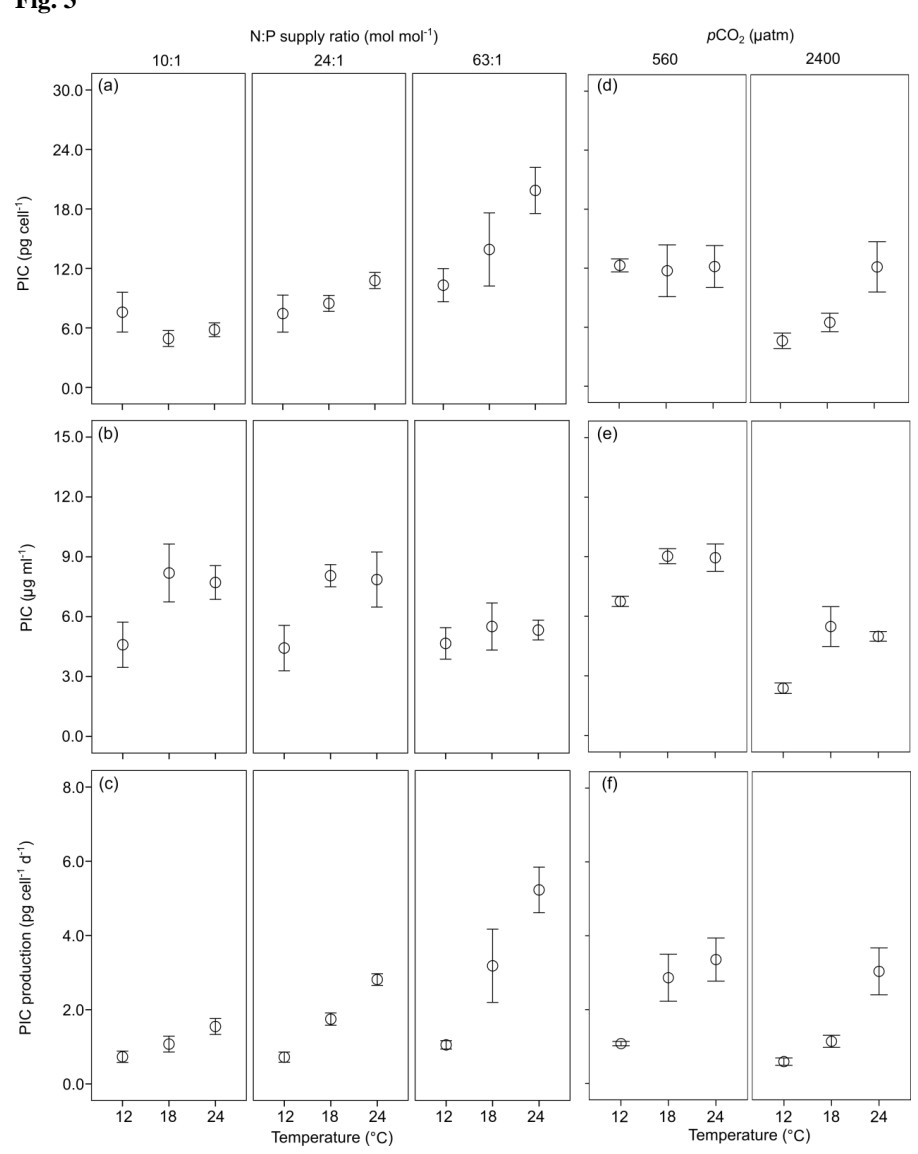










**Fig. 4**

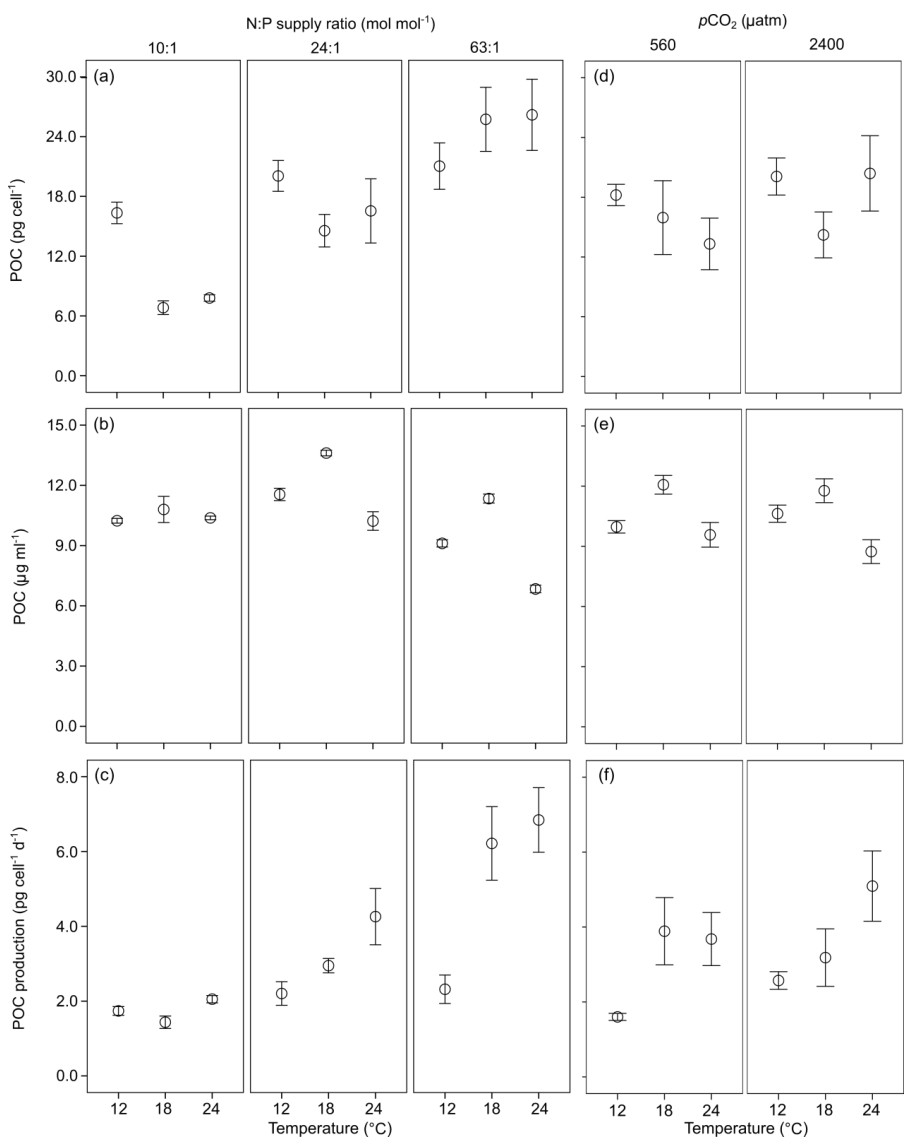








**Fig. 5**

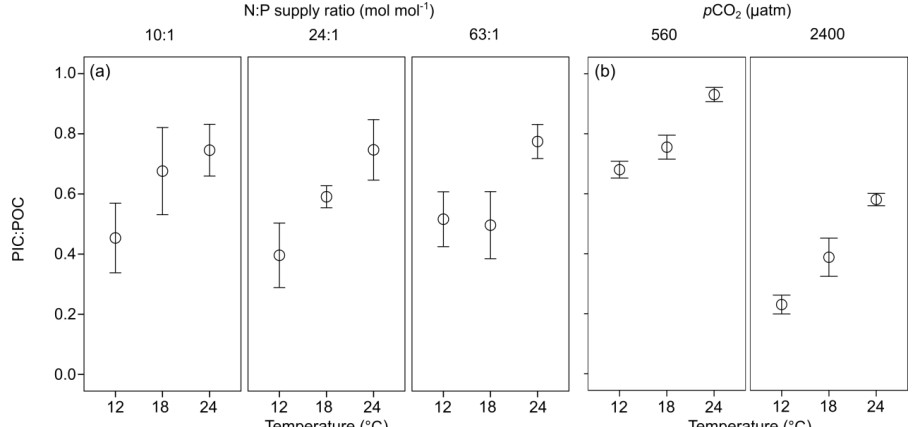




















**Fig. 6**

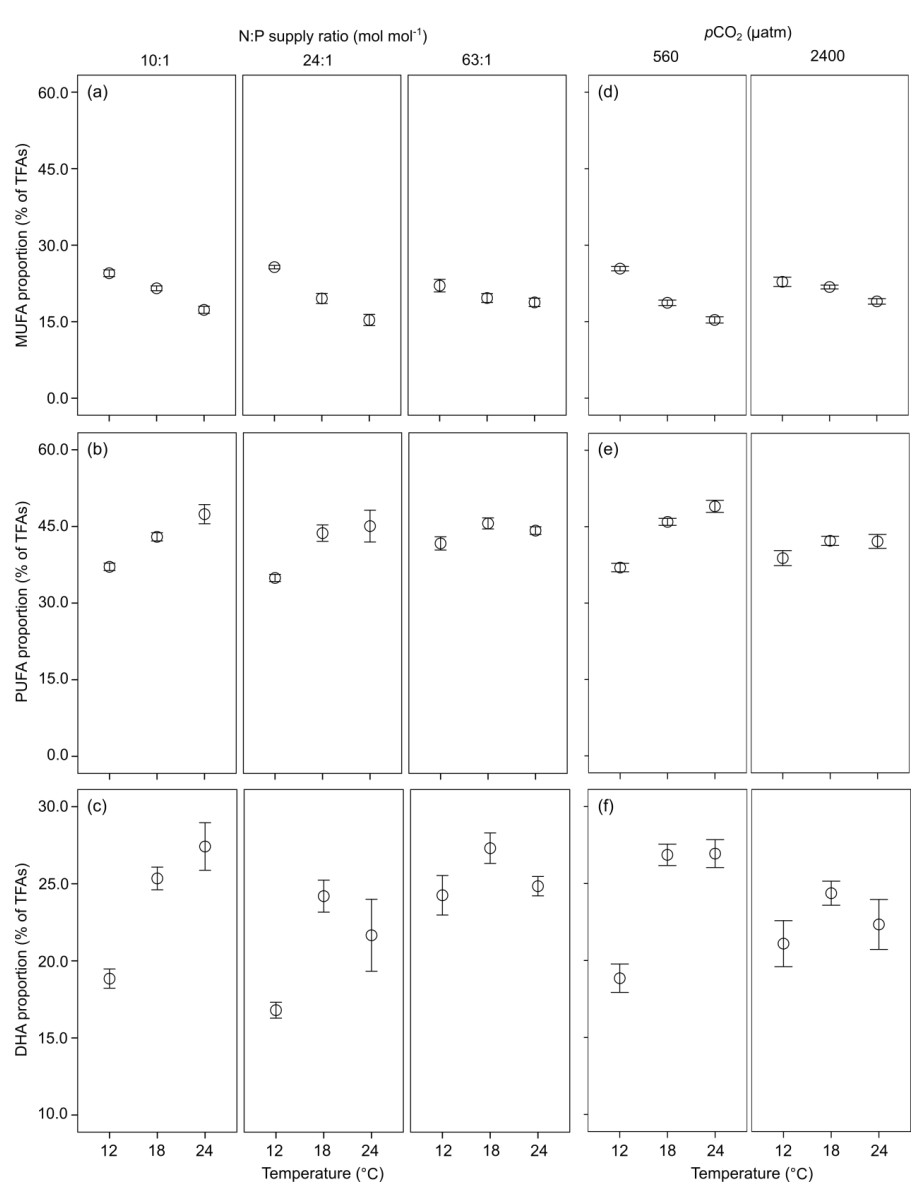







**Fig. 7**

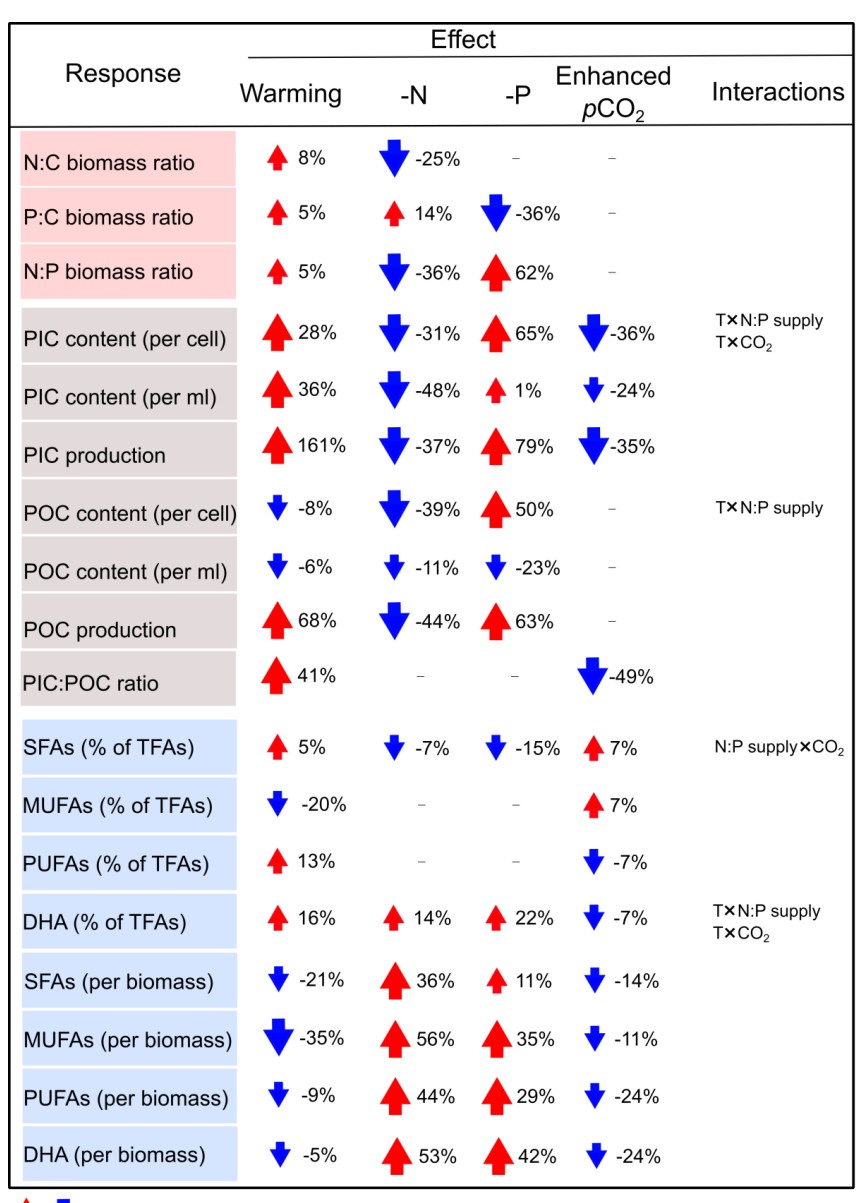