# Peer review of "Simultaneous shifts in elemental stoichiometry and fatty acids of"

_Biogeosciences, 2017_

## Referee Comment (RC1) · Anonymous Referee #1 · 7 Jun 2017

General commonts: This study is an important step in understanding the interactive effects of environmental factors on coccolithophores. The paper has been well written. My general comments are as follows:

Line 30: "PIC" for the first appearance, should be marked it's the abbreviation of "particulate inorganic carbon". Also for "POC". Line 31-32: "10:1, 24:1 and 63:1" are the ratios of N:P, the unite "mol mol-1" , not necessarily shown. Line 87-92: "E.huxleyi is expanding its range poleward", why then gave an example of the subtropical area. Line 149-151: "The target values were chosen to reflect a present and future regime of each factor", however, the pCO2 concentrations 560 and 2400 $\mu$atm they used, can hardly

[Figure]

be considered reasonable. An explanation why a gap in the CO2 concentrations was so big. Line 172: Can they write in detail about how "the specific growth rate of 20% of $\mu$max was applied". I'm curious and puzzled about the reason and methods of how the 20% of $\mu$max ($\mu$) was realized. Usually, specific growth rate is not expressed by %. Line175-176: They said that the incubation water was exchanged with fresh seawater, since the culture medium was partially renewed according to the renewal rate D, the N:P ratios might deviate the target supply ratios in the remained medium due to differential consumption of N and P, can they give some information to show that the N:P supply ratios are stable after several rounds of renewal. Line 178: It seems that the cell concentration was extremely high, the cell concentration range should be provided. Line 180: What do the authors mean by "the net growth rate (r)", what's the difference between r and $\mu$? Confusing wordings or mis-understood definations? Line 203: Here "was" should be "were". Line 241: Is this theory applicable in all species and in any conditions. Line1103: Why there is no panel for the pCO2 effect in Fig. 2. Line 1112: As I read from the "experimental setup" part, this study investigates the combined effects of temperature, pCO2 and N:P supply ratios on E.huxleyi. Why in Fig 3. the combined effects of N:P supply ratio and pCO2 are not considered, i.e. pCO2 is not considered in panel (a), (b), (c), and N:P supply ratio is not considered in panel (d), (e) and (f). The same question for Fig. 4, 5 and 6.

Major revision is needed

General comments:

This study is an important step in understanding the interactive effects of environmental factors on coccolithophores. The paper has been well written. My general comments are as follows:

Line 30: "PIC" for the first appearance, should be marked it's the abbreviation of "particulate inorganic carbon". Also for "POC".

Line 31-32: "10:1, 24:1 and 63:1" are the ratios of N:P, the unite "mol mol$^{-1}$", not necessarily shown.

Line 87-92: "*E.huxleyi* is expanding its range poleward", why then gave an example of the subtropical area.

Line 149-151: "The target values were chosen to reflect a present and future regime of each factor", however, the $pCO_2$ concentrations 560 and 2400 μatm they used, can hardly be considered reasonable. An explanation why a gap in the CO2 concentrations was so big.

Line 172: Can they write in detail about how "the specific growth rate of 20% of $\mu_{max}$ was applied". I'm curious and puzzled about the reason and methods of how the 20% of $\mu_{max}$ ($\mu$) was realized. Usually, specific growth rate is not expressed by %.

Line175-176: They said that the incubation water was exchanged with fresh seawater, since the culture medium was partially renewed according to the renewal rate D, the N:P ratios might deviate the target supply ratios in the remained medium due to differential consumption of N and P, can they give some information to show that the N:P supply ratios are stable after several rounds of renewal.

Line 178: It seems that the cell concentration was extremely high, the cell concentration range should be provided.

Line 180: What do the authors mean by "the net growth rate (r)", what's the difference between r and μ? Confusing wordings or mis-understood definations?

Line 203: Here "was" should be "were".

Line 241: Is this theory applicable in all species and in any conditions.

Line1103: Why there is no panel for the $pCO_2$ effect in Fig. 2.

**Fig. 1.**

---

## Referee Comment (RC2) · Anonymous Referee #2 · 7 Aug 2017

General comments:

First of all, I want to compliment the authors on their work. There aren't many studies out there that manipulate three climate change related factors in a full-factorial design. The experiment is conducted nicely, the results are presented (mostly) clear and the data behind this study is very valuable in understanding the interaction effects of temperature, $CO_2$ and nutrient conditions on phytoplankton physiology. Furthermore, the results obtained from this study are well imbedded in our current knowledge on these topics in the discussion. My main concerns with the manuscript in its current form are the framing of the experimental manipulations to global patterns and the length of the

discussion. There seems to be a mismatch between projected temperature and $CO_2$ conditions in future oceans and the ones you manipulated. I would like to know how you would translate your results to a future ocean scenario as the $CO_2$ concentrations in your lowest treatment are higher than currently measured in global oceans (max 440 ppm; Bakker et al. (2016)). On a similar note, how do the three temperature treatments with a difference of 12C relate to future ocean projections? The discussion is quite lengthy and would benefit in my opinion to focus more on the interaction effects observed in the study as these are the core strength of the work and could advance the field. Perhaps you could reduce the amount of wording if you first discuss the solo effects and then go into all interaction effects in one paragraph (for C:N:P stoichiometry, PIC:POC separately). It seems that there is currently a lot of overlap in the things discussed in separate paragraphs. In addition, I'm missing the inclusion of the PON and POP contents underlying the responses in C:N:P stoichiometry, the results of N:P supply ratio on maximal growth rate and the (though non-significant) results of C:N:P stoichiometry in the different $CO_2$ treatments (Fig. 2). From the introduction it is not clear that PIC and POC production will be discussed. In my opinion, the focus on C:N:P stoichiometry and underlying biochemical composition is the core of your work and introduced very well in the manuscript. I understand the importance of PIC:POC for calcifiers specifically, but I would advise to focus less on the PIC and POC contents, production rates and population yields and more on the C:N:P and fatty acids. A discussion on how changes in stoichiometry and fatty acids relate to each other would be a great addition to the discussion section. What would be a great addition to the introduction are hypotheses on how temperature, $CO_2$ and nutrient supply affect the C:N:P stoichiometry and fatty acid composition. Something similar to Figure 7, but then hypothetical. This would then furthermore help shape the discussion as you could refer back to these hypotheses. A smaller comment, but the use of N:C and P:C ratios instead of C:N and C:P is not very commonly used in literature. The readability and comparison of these ratios to other studies would benefit greatly if they are expressed in C:N and C:P. Furthermore, the reasoning behind the statistical methods used are

not entirely clear for me. Some information in the method section on why this type of statistics are used and what the associated parameters mean would aid the reader in the understanding of the manuscript. All in all, I think this is very interesting work that, with adjustments, would be very suitable for publication in Biogeosciences.

Specific comments:

73 Do you mean community structure of phytoplankton? 80-82 'via releasing $CO_2$' is not really clear for me what this means and why coccolithophores are important components of the carbon cycle. 112 What do you mean by a core feature? 'Element' –> 'elemental' 117 Food for which organism? Phytoplankton or zooplankton? 129-136 This comes as a surprise for me here and seems to fit better in the methodological section than in the introduction 138 PIC:POC is already a ratio 147 The manipulated $CO_2$ levels came as a surprise to me in the framework of current and future projections. Do you have specific reasons to choose these levels as I would have expected a lower 'ambient' $CO_2$ level (around 400 ppm)? 148 Does the strain have a specific reference number (to make possible comparisons with other studies easier)? 175 does 'fresh' seawater imply that is was taken from sea at that day? 222 Is there a specific reason why you choose GLMM's instead of the more classic ANOVA's? 228 What are link functions? 222-239 This part of the statistics is quite difficult for me to follow. Could you explain a bit more about the different procedures and what they do? 242 I would not assume mumax to be the same between nutrient treatments as that was the case in another study. Did you test this and was this the case in your study? 244 Is there a specific reason why you only used w2 for the mumax results and not for other results as for instance figure 7? 248 Why would you use nested models when you have a full factorial design? In other words, what is the added value of these statistical tests? Can you relate your chosen temperatures to acclimatization of E.hux in your lab or the original population that was sampled? How are average annual water temperatures at the Azoren? 274 Did you determine the $CO_2$ effects by post-hoc tests? As there was no overall effect of $CO_2$ on maximum growth rate while you have a significant

interaction effect, wouldn't that mean that the effect of temperature is dependent on the $CO_2$ level, but not vice versa? 280 any particular reason to use N:C ratios as opposed to C:N ratios? The latter is used more often in literature and makes the comparison with the Redfield Ratio easier. For instance, a hump-shaped curve to temperature (or U-shaped curve, line 286) is also observed for a marine cyanobacterium (Fu et al. 2014). By having the ratios in N:C instead of C:N, comparison with other studies like these can get confusing. Furthermore, you did not report interaction effects of temperature and N:P supply ratio on N:P ratios. So how does the difference in temperature response under N and P deficiency (lines 287-288) relate to that? 283 instead of biomass ratios, would it make sense to use PON:POP or POC:PON as that would already imply that it is biomass related. Related to that question, is the C:N ratio composed of TPC:PON or POC:PON? Furthermore, what is underlying the changes in stoichiometry? You have the results for POC content in Figure 4, but how do PON and POP change? 292 What is a PIC population yield? 347 Technically, C:N:P is not a ratio but is composed of C:N and C:P ratios. Additionally, why did you chose to only highlight the N:P results? 356 These interactions effects don't become clear from table 2, as there you only report the effects of the individual stressors. 369 'strains' instead of 'strain' 370 It would be interesting to link this result with the origin of your strain. Does it fall in excepted patterns? 375 I could also argue it the other way, that the biogeographic origin of an E. huxleyi strain is important for their response to temperature. Like mentioned before, could you elaborate on this more? 378 Seems to contrast Table 1 and lines, were you show no effect of $CO_2$ on maximal growth rate. Or is this based on post-hoc comparisons? 387-389 Can you quantify these slopes as they come a bit as a surprise at this point in the manuscript. 393 remove 'and' 394 If it is a conceptual graph you're referring to, I would be interested in the conceptual reasoning behind this response. 403 I would opt for 'C:N:P stoichiometry' instead of biomass ratios. 409 What do you mean by 'prevailed the governing effect'? 415-417 I really like Figure 7 as it gives a nice overview about your results. But what I'm missing there is the change in cellular N and P content. These results could help you in making conclusions about the changes

in PON:POP, whether that is mainly due to N or P deficiency. Furthermore, should it be a table instead of a figure? 445-447 But given your result that the changes in N:C (or C:N) are stronger than those of P:C, what would be the mechanism behind that? Is there any current literature on that respect? Furthermore, if you bring in the argument of less P rich ribosomes with warming, wouldn't you have expected an decrease in P:C instead of the increase you observed? 461 I'm missing here a coupling to your own experimental set-up, did you not find effects of $CO_2$ on stoichiometry due to light conditions or nutrient loads? Can you compare your set-up with those from the studies you mentioned? 462 This is a rather fast transition for me from stoichiometry to cellular biomass. Perhaps this part fits better with the discussion paragraph on growth rates. 472 But you haven't looked at taxonomic composition as you study one species. As there is already such variability between strains and experiments with E.hux, I would shorten this paragraph and focus more on the drivers of variation in responses. 492 Refrain from starting a sentence with 'and' 498 This is vague for me, what other environmental drivers do you mean specifically? 507 'and the present study' should be within brackets? 519 $CO_2$ would not be related to future oceans as the lowest treatment is already elevated. 524 This argument is not clear to me and does not follow logically from your work. Yes, you have changes in PIC and POC yields with environmental changes, but why would that not scale up to carbon export? 529 'dynamic' –> 'dynamics' 595 'low trophic levels consumers': do you mean first order consumers? 606 'relationship' –> 'relationships' 612 How does the temperature and $CO_2$ relate to future ocean scenarios? That would be good to add to the introduction. 614 Wouldn't that contradict the argument you made in line 523-524 that these results cannot be scaled up to carbon export?

Comments on figures and tables:

Table S2: the meaning of the column effect builder is not clear to me. What does main, two way and three way mean and how do these model outputs relate to the ones in table 2? Table 2: It is not clear to me what a significant intercept in these models mean?

Furthermore, I'm missing interaction terms for some of the variables. I would change PIC (ug/ml) to PIC population yield (ug/ml) to make it easier to connect with the text. Figure 1: I'm missing the results for N:P supply in this figure. Figure 2: I'm missing the results for $CO_2$ in this figure. Table S4 seems to be the only results in which standard deviations instead of standard errors are reported. For consistency reasons I would opt for standard errors here. Fig S2 is missing the (mean +/- SE) from the legend. Or is standard deviation expressed here?

References:

Bakker, D. C. E., B. Pfeil, C. S. Landa, N. Metzl, K. M. O'Brien, A. Olsen, K. Smith, C. Cosca, S. Harasawa, S. D. Jones, S. I. Nakaoka, Y. Nojiri, U. Schuster, T. Steinhoff, C. Sweeney, T. Takahashi, B. Tilbrook, C. Wada, R. Wanninkhof, S. R. Alin, C. F. Balestrini, L. Barbero, N. R. Bates, A. A. Bianchi, F. Bonou, J. Boutin, Y. Bozec, E. F. Burger, W. J. Cai, R. D. Castle, L. Chen, M. Chierici, K. Currie, W. Evans, C. Featherstone, R. A. Feely, A. Fransson, C. Goyet, N. Greenwood, L. Gregor, S. Hankin, N. J. Hardman-Mountford, J. Harlay, J. Hauck, M. Hoppema, M. P. Humphreys, C. W. Hunt, B. Huss, J. S. P. Ibánhez, T. Johannessen, R. Keeling, V. Kitidis, A. Körtzinger, A. Kozyr, E. Krasakopoulou, A. Kuwata, P. Landschützer, S. K. Lauvset, N. Lefèvre, C. Lo Monaco, A. Manke, J. T. Mathis, L. Merlivat, F. J. Millero, P. M. S. Monteiro, D. R. Munro, A. Murata, T. Newberger, A. M. Omar, T. Ono, K. Paterson, D. Pearce, D. Pierrot, L. L. Robbins, S. Saito, J. Salisbury, R. Schlitzer, B. Schneider, R. Schweitzer, R. Sieger, I. Skjelvan, K. F. Sullivan, S. C. Sutherland, A. J. Sutton, K. Tadokoro, M. Telszewski, M. Tuma, S. M. A. C. van Heuven, D. Vandemark, B. Ward, A. J. Watson, and S. Xu. 2016. A multi-decade record of high-quality $fCO_2$ data in version 3 of the Surface Ocean $CO_2$ Atlas (SOCAT). Earth Syst. Sci. Data 8:383-413.

Fu, F.-X., E. Yu, N. S. Garcia, J. Gale, Y. Luo, E. A. Webb, and D. A. Hutchins. 2014. Differing responses of marine N-2 fixers to warming and consequences for future diazotroph community structure. Aquatic Microbial Ecology 72:33-46.

---

## Author Comment (AC1) · 30 Aug 2017

Responses to comments from Reviewer 1

Comment 1 Line 30: "PIC" for the first appearance, should be marked it's the abbreviation of "particulate inorganic carbon". Also for "POC".

Response: As suggested, the abbreviation 'particulate inorganic carbon' for PIC and 'particulate organic carbon' for POC will be added for their first appearance.

Comment 2 Line 31-32: "10:1, 24:1 and 63:1" are the ratios of N:P, the unite "mol mol-1", not necessarily shown.

[Figure]

Response: As suggested, we will remove the unit 'mol mol-1' and state 'molar ratios 10:1, 24:1 and 63:1'.

Comment 3 Line 87-92: "E.huxleyi is expanding its range poleward", why then gave an example of the subtropical area.

Response: The study in the subtropical area was removed. A study in the Bering Sea will be added, in which Harada et al. (2012) found that warming and freshening have promoted Emiliania huxleyi blooms in the Bering Sea since the late 1970s.

Comment 4 L149-151: "The target values were chosen to reflect a present and future regime of each factor", however, the pCO2 concentrations 560 and 2400 $\mu$atm they used, can hardly be considered reasonable. An explanation why a gap in the CO2 concentrations was so big.

Response: In plankton-rich waters respiration plus atmospheric CO2-enrichment can drive regional pCO2 up to 900 $\mu$atm at times today. Considerable seasonal, depth and regional variations of pCO2 have been observed in the present-day ocean (Joint et al. 2011). For example, up to 900 $\mu$atm of pCO2 was observed in August in the Southern Bight of the North Sea, with a lower pCO2 (192 $\mu$atm) in April (Schiettecatte et al. 2007). A natural pCO2 gradient of 292 to 8828 $\mu$atm was reported off Culcano Island, Italy (Ziveri et al. 2014). In the future oceans, pCO2 will increase with rising atmospheric CO2, being 851-1370 $\mu$atm by 2100 and 1371-2900 $\mu$atm by 2150 (RCP8.5 scenario of the IPCC report 2014) (IPCC 2014). In the present study, the chosen values of pCO2 cover the range of typical levels of pCO2 in the present-day ocean and future ocean projections. Such a big gap in the value of pCO2 was used to test the response of E. huxleyi to a considerable variation of pCO2, which has been observed in the present-day ocean as mentioned above. To clarify the reason of pCO2 set-up in our study, a detailed explanation will be added in the revised manuscript.

Comment 5 Line 172: Can they write in detail about how "the specific growth rate of 20% of $\mu$max was applied". I'm curious and puzzled about the reason and methods of

how the 20% of $\mu$max ($\mu$) was realized. Usually, specific growth rate is not expressed by %.

Response: Using % of $\mu$max guarantees that the strength on nutrient deficiency is equal through all temperature and pCO2 treatments. A fixed value of $\mu$ would mean weak deficiency when $\mu$max is low and strong deficiency when it is high. Based on the specific growth rate ($\mu$= 20% of $\mu$max (day-1)), the equivalent daily renewal rate (D, day-1) can be estimated according to the equation D = 1- e-$\mu$t, where t is renewal interval (day) (here t = 1 day). Thus, the volume of the daily renewal incubation water can be calculated by multiplying D with the total volume of incubation water. We will provide the detail about how 20% of $\mu$max was realized and applied and the reason of using % of $\mu$max in the revised manuscript.

Comment 6 Line175-176: They said that the incubation water was exchanged with fresh seawater, since the culture medium was partially renewed according to the renewal rate D, the N:P ratios might deviate the target supply ratios in the remained medium due to differential consumption of N and P, can they give some information to show that the N:P supply ratios are stable after several rounds of renewal.

Response: 'fresh' seawater here implies freshly made seawater medium with the target N:P supply ratio, but not only fresh seawater. Indeed, nutrient concentrations in semi-continuous culture may deviate from the target values due to consumption. Semi-continuous cultures, as a practical surrogate for fully continuous culture, have been successfully used to study the effect of nutrients on phytoplankton stoichiometry and fatty acid composition (Terry et al. 1985; Lynn et al. 2000; Piepho et al. 2012; Feng et al. 2017). While we did not measure the N and P in the media daily, semi-continuous cultures can be applied to study the effect of N:P supply ratio on E. huxleyi stoichiometry and fatty acid composition. We will clarify that the incubation water was exchanged with freshly made seawater medium in the revised manuscript.

Comment 7 Line 178: It seems that the cell concentration was extremely high, the cell

concentration range should be provided.

Response: In our study, the final cell density at the steady state ranged between $1.50 \times 105 - 17.8 \times 105$ cells mL-1, with the average value of $7.95 \times 105$ cells mL-1. High cell density ($> 1 \times 106$ cells mL-1) was observed in six out of 18 treatments. The average value of cell density in our study was consistent with the range of those in previous studies. For example, De Bodt et al. (2010) reported the maximum cell density of $6.84 \times 105$ cells mL-1 when testing the effects of $pCO_2$ and temperature on E. huxleyi calcification. As suggested, the range of cell densities will be shown in the revised manuscript.

Comment 8 Line 180: What do the authors mean by "the net growth rate (r)", what's the difference between r and $\mu$? Confusing wordings or mis-understood definations?

Response: In order to clarify the difference between $\mu$ and r, we will use the term 'gross growth rate' for $\mu$, while r (net growth rate) is the difference between the gross growth rate and the loss rate ($r = \mu - D$). The difference between gross growth rate and net growth rate will be clarified in the revised manuscript.

Comment 9 Line 203: Here "was" should be "were".

Response: As suggested, the word 'was' will be replaced by 'were'.

Comment 10 Line 241: Is this theory applicable in all species and in any conditions.

Response: This hypothesis was proposed by Cherif and Loreau (2010), suggesting that realized maximum growth rates (i.e., the observed maximum growth rate in the present study, $\mu_{max}$) should be equal for essential, non-substantial resources for phytoplankton species. This assumption was supported by both theoretical and empirical evidence, 1) lab experiments showed little or no luxury uptake of resources at the highest growth rate; 2) the maximum capacity of the uptake machinery should not be oversized for a given resource based on economical design (Cherif and Loreau 2010). Similar $\mu_{max}$ in different nutrient conditions has been observed for different phytoplankton species in empirical experiments (e.g., Ahlgren 1985; Baek et al. 2008; Bi et al. 2012). The model presented in Cherif and Loreau (2010) has also been successfully used to study how nutrient gradients influence stoichiometry of autotrophs in natural chemostats (Nifong et al. 2014). In the present study, we had only one value of $\mu$max for each nutrient treatment under different temperature and pCO2 conditions, thus the effect of N:P supply ratio cannot be tested with ANOVA efficiently. In the revised manuscript, we tested the response of $\mu$max to temperature, N:P supply ratio and pCO2 using GLMMs. The results of GLMMs were consistent with those of ANOVA, showing a highly significant effect of temperature on $\mu$max. As the chosen best model contained only first order effects, no significant interactions between the three environmental factors were detected. The non-significant response of $\mu$max to N:P supply ratio in E. huxleyi is consistent with the assumption of Cherif and Loreau (2010). We will revise the Results and Discussion sections according to the new results of GLMMs on $\mu$max.

Comment 11 Line1103: Why there is no panel for the pCO2 effect in Fig.2.

Response: The effect of pCO2 on stoichiometric C:N:P will be added in Fig. 2.

Comment 12 Line 1112: As I read from the "experimental setup" part, this study investigates the combined effects of temperature, pCO2 and N:P supply ratios on E.huxleyi. Why in Fig 3. the combined effects of N:P supply ratio and pCO2 are not considered, i.e. pCO2 is not considered in panel (a), (b), (c), and N:P supply ratio is not considered in panel (d), (e) and (f). The same question for Fig. 4, 5 and 6.?

Response: In the present study, significant interactions between temperature and N:P supply ratio, and between temperature and pCO2 were detected for cellular PIC and POC contents, and the proportion of DHA. However, the significant interaction between N:P supply ratio and pCO2 was only found for the proportion of SFAs. Please see Fig. 7 for a systematic summary. Thus, the interactions between temperature and N:P supply ratio, and between temperature and pCO2 were shown in figures 3-6, while

that between N:P supply ratio and pCO2 was only shown for SFAs in the supporting information Fig. S2.

Responses to comments from Reviewer 2

General comment I My main concerns with the manuscript in its current form are the framing of the experimental manipulations to global patterns and the length of the discussion. There seems to be a mismatch between projected temperature and CO2 conditions in future oceans and the ones you manipulated. I would like to know how you would translate your results to a future ocean scenario as the CO2 concentrations in your lowest treatment are higher than currently measured in global oceans (max 440 ppm; Bakker et al. (2016)). On a similar note, how do the three temperature treatments with a difference of 12C relate to future ocean projections?

Response: The chosen levels of pCO2 and temperature in this study were set based on the reasons below: 1) pCO2. Please see our reply to Reviewer 1 Comment 4. In plankton rich waters respiration plus atmospheric CO2-enrichment can drive regional pCO2 up to 900 $\mu$atm at times today. For example, up to 900 $\mu$atm of pCO2 was observed in August in the Southern Bight of the North Sea (Schiettecatte et al. 2007). A much higher pCO2 (a natural pCO2 gradient of 292 to 8828 $\mu$atm) was observed off Culcano Island, Italy (Ziveri et al. 2014). In the future oceans, pCO2 will increase with rising atmospheric CO2, being 851-1370 $\mu$atm by 2100 and 1371-2900 $\mu$atm by 2150 (RCP8.5 scenario of the IPCC report 2014) (IPCC 2014). Therefore, the chosen values of pCO2 in the present study cover the range of typical levels of pCO2 in the present-day ocean and future ocean projections. 2) Temperature. Water surface temperatures at the Azores vary between $\sim$12 - 29°C (Lafon et al. 2004), with the inter-annual average temperature between 16 - 22°C and peaks usually reaching a maximum of 24 - 25°C (http://dive.visitazores.com/en/when-dive; last accessed date: 22.08.2017). Our temperature range setup was based on the study of Lewandowska et al. (2014), who chose a temperature increment of 6°C, according to the ocean general circulation model under the IPCC SRES A1F1 scenario. Annual mean sea surface temperature

across the North Atlantic (0–60° N) is projected to reach 29.8 °C in 2100 according to the ocean general circulation model (Lewandowska et al. 2014). We also chose this setup to compare with our previous results (Bi et al. 2017). 3) The ranges of pCO2 and temperature in our study is the same with those in our previous work (Bi et al. 2017), which makes the comparison easier between different work. The reason of pCO2 and temperature setup will be pointed out in Material and Method in the revised manuscript. In the Introduction, we will also explain how the setup of temperature and pCO2 in our study relates to future ocean scenarios.

General comment II The discussion is quite lengthy and would benefit in my opinion to focus more on the interaction effects observed in the study as these are the core strength of the work and could advance the field. Perhaps you could reduce the amount of wording if you first discuss the solo effects and then go into all interaction effects in one paragraph (for C:N:P stoichiometry, PIC:POC separately). It seems that there is currently a lot of overlap in the things discussed in separate paragraphs.

Response: As suggested, we will first discuss the solo effect and then go into all interaction in the Discussion.

General comment III In addition, I'm missing the inclusion of the PON and POP contents underlying the responses in C:N:P stoichiometry, the results of N:P supply ratio on maximal growth rate and the (though non-significant) results of C:N:P stoichiometry in the different CO2 treatments (Fig. 2).

Response: As suggested, the results of PON and POP contents, the N:P effect on $\mu$max in Fig. 1, and the pCO2 effect on C:N:P stoichiometry in Fig. 2 will be shown in the revised manuscript.

General comment IV From the introduction it is not clear that PIC and POC production will be discussed. In my opinion, the focus on C:N:P stoichiometry and underlying biochemical composition is the core of your work and introduced very well in the manuscript. I understand the importance of PIC:POC for calcifiers specifically, but I

would advise to focus less on the PIC and POC contents, production rates and population yields and more on the C:N:P and fatty acids.

Response: We will shorten the results and discussion on PIC and POC and focus more on the changes in C:N:P stoichiometry and fatty acids.

General comment V A discussion on how changes in stoichiometry and fatty acids relate to each other would be a great addition to the discussion section.

Response: A discussion on how the changes in stoichiometry relate to those in fatty acids will be added.

General comment VI What would be a great addition to the introduction are hypotheses on how temperature, CO2 and nutrient supply affect the C:N:P stoichiometry and fatty acid composition. Something similar to Figure 7, but then hypothetical. This would then furthermore help shape the discussion as you could refer back to these hypotheses.

Response: In the last paragraph of the introduction, we will add hypotheses on how temperature, pCO2 and nutrient supply affect C:N:P stoichiometry and fatty acid composition.

General comment VII A smaller comment, but the use of N:C and P:C ratios instead of C:N and C:P is not very commonly used in literature. The readability and comparison of these ratios to other studies would benefit greatly if they are expressed in C:N and C:P.

Response: We will use POC:PON and POC:POP, instead of N:C and P:C biomass ratios, in the revised manuscript.

General comment VIII Furthermore, the reasoning behind the statistical methods used are not entirely clear for me. Some information in the method section on why this type of statistics are used and what the associated parameters mean would aid the reader in the understanding of the manuscript.

Response: Generalized linear mixed models (GLMMs) are appropriate for non-normal data such as counts or proportions, while classical statistical procedures such as ANOVA rely on normally distributed data (Bolker et al. 2009). GLMMs combine the properties of two statistical models (linear mixed models and generalized linear models) (Bolker et al. 2009) and have been widely used in ecology (e.g., Frère et al. 2010; Jamil et al. 2014; Bracewell et al. 2017), in which data sets are often non-normally distributed. We will explain the reason why to choose GLMMs and what the associated parameters mean in the revised manuscript.

Specific comment 1 73 Do you mean community structure of phytoplankton?

Response: According to Doney et al. (2012), climate change may alter the physiological functioning, behavior, and demographic traits of organisms. These changes cascade from primary producers to upper trophic levels such as fish, seabirds and marine mammals. Therefore, community structure in the sentence in our manuscript means not only for phytoplankton but also for other trophic levels. We hence will clarify this as '—- community structure of different trophic levels —'.

Specific comment 2 80-82 'via releasing CO2' is not really clear for me what this means and why coccolithophores are important components of the carbon cycle.

Response: This sentence will be revised to clarify that coccolithophores are not only important photosynthetic producers of organic matters (causing a draw-down of CO2 in the surface layer), but also play predominant roles in the production and export of calcium carbonate to deeper layers (causing a net release of CO2 to the atmosphere).

Specific comment 3 112 What do you mean by a core feature? 'Element' –> 'elemental'

Response: This sentence will be revised to ' — variability in Emiliania huxleyi C:N:P stoichiometry (cellular quotas and ratios of C, N and P) can also be important in ocean biogeochemistry.' 'element' will be changed to 'elemental'.

Specific comment 4 117 Food for which organism? Phytoplankton or zooplankton?

Response: According to Rosenblatt and Schmitz (2016), shifts in resource nutrient content are generally occur with shifts in consumer physiology and behavior, and they are often overlooked in studies of the responses of food web dynamics to climate change. We thus will clarify this sentence as ' — shifts in resource nutrient content for consumers are often overlooked in climate change ecology –.'.

Specific comment 5 129-136 This comes as a surprise for me here and seems to fit better in the methodological section than in the introduction.

Response: As suggested, these two sentences will be moved to the methodological section.

Specific comment 6 138 PIC:POC is already a ratio

Response: It will be revised as 'PIC and POC contents and their ratios' throughout the text.

Specific comment 7 147 The manipulated $CO_2$ levels came as a surprise to me in the framework of current and future projections. Do you have specific reasons to choose these levels as I would have expected a lower 'ambient' $CO_2$ level (around 400 ppm)?

Response: Please see our reply to General comment I.

Specific comment 8 148 Does the strain have a specific reference number (to make possible comparisons with other studies easier)?

Response: As suggested, the specific reference number (internal culture collection reference code: A8) will be added.

Specific comment 9 175 does 'fresh' seawater imply that is was taken from sea at that day?

Response: 'fresh' seawater implies that freshly made seawater medium, but the seawater was not taken from the sea on that day. To clarify this, the sentence will be revised as 'The incubation water was exchanged with freshly made seawater medium

—'.

Specific comment 10 222 Is there a specific reason why you choose GLMM's instead of the more classic ANOVA's?

Response: Please see our reply to General comment VIII.

Specific comment 11 228 What are link functions?

Response: The link function is a transformation of the target that allows estimation of the model (https://www.ibm.com/support/knowledgecenter/SSLVMB_21.0.0/com.ibm.spss.statistics.help/idh_glmm_target.htm; last accessed date: 14.08.2017). For example, identity link function is appropriate with any distribution except for multinomial, while logit can be used only with the binomial or multinomial distribution. We will explain in the text what the link function is.

Specific comment 12 222-239 This part of the statistics is quite difficult for me to follow. Could you explain a bit more about the different procedures and what they do?

Response: We will explain more about the different procedures in GLMMs.

Specific comment 13 242 I would not assume mumax to be the same between nutrient treatments as that was the case in another study. Did you test this and was this the case in your study?

Response: We tested the changes of $\mu$max between different nutrient treatments. Because there was only one data of $\mu$max in each nutrient treatment under different temperature and $pCO2$ conditions, the effect of N:P supply ratio cannot be tested with ANOVA efficiently. In the revised manuscript, we will show the results of GLMMs on the response of $\mu$max to temperature, N:P supply ratio and $pCO2$ using GLMMs, which are consistent with those of ANOVA, showing a highly significant effect of temperature and non-significant effect of N:P supply ratio and $pCO2$. As the chosen best model contained only first order effects, no significant interactions between the three environmental factors were detected. In the literature, there are limited data on the response of

$\mu$max in E. huxleyi to nutrient availability, while several studies reported the response of specific growth rate. According to Cherif and Loreau (2010), realized maximum growth rates (i.e., the observed maximum growth rate in the present study, $\mu$max) should be equal for essential, non-substantial resources for phytoplankton species. This assumption was supported by both theoretical and empirical evidence, 1) lab experiments showed little or no luxury uptake of resources at the highest growth rate; 2) the maximum capacity of the uptake machinery should not be oversized for a given resource based on economical design (Cherif and Loreau 2010). For E. huxleyi, luxury consumptions for phosphate and nitrate are lower than other phytoplankton taxa (Rost and Riebesell 2004). Thus, the non-significant response of $\mu$max to N:P supply ratio in E. huxleyi in our study is consistent with the assumption of Cherif and Loreau (2010). Future work is suggested to study the response of $\mu$max in E. huxleyi under a wider range of nutrient conditions. We will revise the Results and Discussion sections according to the new results of GLMMs on $\mu$max.

Specific comment 14 244 Is there a specific reason why you only used w2 for the mumax results and not for other results as for instance figure 7?

Response: It would be better to show w2 for all responses; however, error mean square cannot be obtained from GLMMs and thus w2 cannot be calculated for response variables tested with GLMMs. In the revised manuscript, we will show the response of $\mu$max using GLMMs. Thus, the results of w2 will be removed and percent changes of $\mu$max will be calculated.

Specific comment 15 248 Why would you use nested models when you have a full factorial design? In other words, what is the added value of these statistical tests? Can you relate your chosen temperatures to acclimatization of E.hux in your lab or the original population that was sampled? How are average annual water temperatures at the Azoren?

Response: It is possible to use a nested model in a full-factorial design setting. The

question a nested model addresses is, whether one factor plays a role under one (or several) configuration(s) of another factor, but not under all configurations of that factor equally. The difference to e.g. a test including straight-forward interaction effects is that interaction terms describe systematic variation of one factor's effects over a gradient of the other, whereas a nested model can highlight if for example pCO2 plays a role for fatty acid content only at intermediate temperature. Please see our reply to General comment I regarding average water surface temperatures. The chosen temperature setup in our study is within the range of sea surface temperature at the Azores.

Specific comment 16 274 Did you determine the CO2 effects by post-hoc tests? As there was no overall effect of CO2 on maximum growth rate while you have a significant interaction effect, wouldn't that mean that the effect of temperature is dependent on the CO2 level, but not vice versa?

Response: In our study, a post hoc test was applied only if there were significant effects. We thus did not determine the effect of pCO2 by the post hoc test, as the effect of pCO2 was not significant according to ANOVA. We agree with the reviewer that the effect of temperature is dependent on the CO2 level. In the revised manuscript, we will use GLMMs to test the response of $\mu$max (Please see our response to Specific comment 13). The results showed no significant interactions between temperature and pCO2, while there was still a different trend of $\mu$max to increase with increasing temperature between the two pCO2 treatments. We will revise the Results and Discussion sections accordingly.

Specific comment 17 280 any particular reason to use N:C ratios as opposed to C:N ratios? The latter is used more often in literature and makes the comparison with the Redfield Ratio easier. For instance, a hump-shaped curve to temperature (or Ushaped curve, line 286) is also observed for a marine cyanobacterium (Fu et al. 2014). By having the ratios in N:C instead of C:N, comparison with other studies like these can get confusing. Furthermore, you did not report interaction effects of temperature and N:P supply ratio on N:P ratios. So how does the difference in temperature response

under N and P deficiency (lines 287-288) relate to that?

Response: We will present the results of POC:PON and POC:POP in the revised manuscript to make the comparison with the Redfield Ratio and the results in the literature easier. We found that, similar to the results in Fu et al. (2014), a hump-shaped curve to temperature was also observed for POC:PON in response to increasing temperature under N deficiency in our study. Indeed, there was non-significant interaction between temperature and N:P supply ratio on PON:POP according to GLMMs. However, POC:PON responded significantly to temperature, showing a different trend of changes to increasing temperature under different N:P supply ratio. We thus presented this nutrient-dependent response, as it need not be universal to constitute 'significant discovery'.

Specific comment 18 283 instead of biomass ratios, would it make sense to use PON:POP or POC:PON as that would already imply that it is biomass related. Related to that question, is the C:N ratio composed of TPC:PON or POC:PON? Furthermore, what is underlying the changes in stoichiometry? You have the results for POC content in Figure 4, but how do PON and POP change?

Response: As suggested, we will use POC:PON, POC:POP and PON:POP in the revised manuscript. The C:N biomass ratio is composed of POC:PON. To explore what is underlying the changes in C:N:P stoichiometry, we analyzed the responses of cellular PON and POP contents. For example, a U-shaped curve was observed for the responses of cellular POC and PON contents to increasing temperature under N deficiency, which can explain the observed hump-shaped curve for the response of POC:PON. The detail results will be included and discussed in the revised manuscript.

Specific comment 19 292 What is a PIC population yield?

Response: A population yield of PIC is the PIC content per ml ($\mu$g ml-1). This will be clarified in the revised manuscript.

Specific comment 20 347 Technically, C:N:P is not a ratio but is composed of C:N and C:P ratios. Additionally, why did you chose to only highlight the N:P results?

Response: As suggested, C:N:P biomass ratio will be changed to C:N:P stoichiometry throughout the text. The response of N:P biomass ratio was highlighted here because it had the highest percent changes among the three stoichiometric ratios. To clarify this, a 'e.g.' will be added in this sentence.

Specific comment 21 356 These interactions effects don't become clear from table 2, as there you only report the effects of the individual stressors.

Response: Indeed, we did not observe significant effects of the three stressors on all response parameters, with significant interactive effects only observed for cellular POC content, and SFA and DHA proportion. We will clarify this information in the first paragraph in the Discussion.

Specific comment 22 369 'strains' instead of 'strain'

Response: As suggested, the word will be corrected to 'strains'.

Specific comment 23 370 It would be interesting to link this result with the origin of your strain. Does it fall in excepted patterns?

Response: In our study, $\mu$max of E. huxleyi (from the Azores, $\sim 38°$N) was two to three times higher at the highest temperature than that at the lowest temperature, showing a similar change pattern with that in E. huxleyi (1.6 times higher at the higher temperature) from the Sargasso Sea ($\sim 20$-$35°$N). We will add the comparison between our results and the results in the literature in the revised manuscript.

Specific comment 24 375 I could also argue it the other way, that the biogeographic origin of an E. huxleyi strain is important for their response to temperature. Like mentioned before, could you elaborate on this more?

Response: We agree that the results show the importance of the biogeographic origin

of an E. huxleyi strain for their response to temperature. We will revise this sentence according to this comment and Specific comment 23.

Specific comment 25 378 Seems to contrast Table 1 and lines, were you show no effect of $CO_2$ on maximal growth rate. Or is this based on post-hoc comparisons?

Response: Please see our response to Specific comment 16.

Specific comment 26 387-389 Can you quantify these slopes as they come a bit as a surprise at this point in the manuscript.

Response: We will quantify these slopes both in the Result and Discussion.

Specific comment 27 393 remove 'and'

Response: 'and' will be removed.

Specific comment 28 394 If it is a conceptual graph you're referring to, I would be interested in the conceptual reasoning behind this response.

Response: The conceptual reasoning behind is still unclear. The authors who proposed the conceptual graph suggested that one possible explanation is that increasing temperature may modulate the balance between a fertilizing effect of ocean carbonation and a metabolic repression by ocean acidification (Sett et al. 2014). This explanation will be added at the end of this paragraph.

Specific comment 29 403 I would opt for 'C:N:P stoichiometry' instead of biomass ratios.

Response: 'C:N:P biomass ratios' was revised to 'C:N:P stoichiometry' throughout the manuscript.

Specific comment 30 409 What do you mean by 'prevailed the governing effect'?

Response: Skau (2015) tested the effects of temperature and phosphorus on stoichiometry in three haptophytes, showing that phosphorus treatments had a stronger

effect on C:P ratios in E. huxleyi compared to temperature. We will revise this sentence as '— nutrient availability had a stronger effect on stoichiometric ratios in E. huxleyi compared to temperature'.

Specific comment 31 415-417 I really like Figure 7 as it gives a nice overview about your results. But what I'm missing there is the change in cellular N and P content. These results could help you in making conclusions about the changes in PON:POP, whether that is mainly due to N or P deficiency. Furthermore, should it be a table instead of a figure?

Response: We will add the changes in cellular N and P contents. Yes, Fig. 7 should be a table, which will be changed in the revised manuscript.

Specific comment 32 445-447 But given your result that the changes in N:C (or C:N) are stronger than those of P:C, what would be the mechanism behind that? Is there any current literature on that respect? Furthermore, if you bring in the argument of less P rich ribosomes with warming, wouldn't you have expected an decrease in P:C instead of the increase you observed?

Response: In the literature, variable changes of PON:POC and POP:POC to warming were observed in E. huxleyi, showing positive (Feng et al. 2008; Matson et al. 2016), negative (Borchard and Engel 2012) and U-shaped responses (Rosas-Navarro et al. 2016). Similar to our study, Borchard and Engel (2012) also found a stronger change of PON:POC than that of POP:POC at higher P condition, while both biomass ratios decreased with increasing temperature. The mechanism behind the different responses of PON:POC and POP:POC may be explained by the temperature-dependent physiology hypothesis, which shows that organisms in warmer conditions require fewer P-rich ribosomes, relative to N-rich proteins (Toseland et al. 2013). We will revise this part of discussion to clarify the mechanism behind the changes of C:N:P stoichiometry in response to warming.

Specific comment 33 461 I'm missing here a coupling to your own experimental setup, did you not find effects of CO2 on stoichiometry due to light conditions or nutrient loads? Can you compare your set-up with those from the studies you mentioned?

Response: We will add comparison of experimental set-up between our study and previous work. For example, Feng et al. (2008) reported that rising pCO2 caused the increase in POC:PON only at the high light condition (400 $\mu$mol photons m-2 s-1). The light intensity in our study (100 $\mu$mol photons m-2 s-1) was lower than that in Feng et al. (2008). In our study, the effect of light intensity was not tested, while the effect of pCO2 was not found under different N:P supply ratios. Thus, future work is suggested to address the effect of light intensity on the E. huxleyi strain used in our study.

Specific comment 34 462 This is a rather fast transition for me from stoichiometry to cellular biomass. Perhaps this part fits better with the discussion paragraph on growth rates.

Response: In this sentence, we still discuss the responses of C:N:P stoichiometry and not cellular biomass. We will revise this sentence as 'Taken together, our results indicate that the changes in N:P supply ratios are reflected in C:N:P stoichiometry in E. huxleyi, ——'.

Specific comment 35 472 But you haven't looked at taxonomic composition as you study one species. As there is already such variability between strains and experiments with E.hux, I would shorten this paragraph and focus more on the drivers of variation in responses.

Response: As suggested, the discussion in this paragraph will be revised to focus more on the drivers of variation in stoichiometric responses: 'Taken together, our results indicate that the changes in N:P supply ratios are reflected in C:N:P stoichiometry in E. huxleyi, across different temperatures and pCO2 levels, showing the absence of significant interactions between the three environmental factors. However, for two algal species from non-calcifying classes (the diatom P. tricornutum and the cryptophyte Rhodomonas sp.) temperature had the most consistent significant effect on N:C

and P:C biomass ratios and showed significant interactions with N:P supply ratios and $pCO_2$ in our previous work (Bi et al. 2017). The results above are consistent with previous studies, which showed that both temperature and N:P supply ratios were ranked as important factors in regulating phytoplankton stoichiometry in previous studies (Boyd et al. 2010; Feng 2015).'.

Specific comment 36 492 Refrain from starting a sentence with 'and'

Response: 'and' will be removed from the beginning of the sentence.

Specific comment 37 498 This is vague for me, what other environmental drivers do you mean specifically?

Response: According to previous studies, the interaction of $pCO_2$ with other environmental factors such as irradiance and temperature may be potential drivers on the changes in PIC:POC (Feng et al. 2008; De Bodt et al. 2010). We will revise this sentence to specify what other environmental drivers are.

Specific comment 38 507 'and the present study' should be within the brackets?

Response: 'the present study' will be added within brackets.

Specific comment 39 519 $CO_2$ would not be related to future oceans as the lowest treatment is already elevated.

Response: Please see our reply to General comment I.

Specific comment 40 524 This argument is not clear to me and does not follow logically from your work. Yes, you have changes in PIC and POC yields with environmental changes, but why would that not scale up to carbon export?

Response: We will revise this sentence as 'It is worth noting that cellular PIC and POC contents are a measure for physiological response and cannot be directly used to infer population response, as different responses between cellular and population yields of PIC (and POC) to environmental changes were evident in previous work (Matthiessen

et al. 2012) and the present study (Fig. 7). Thus scaling our results up to coccolithophores carbon export should consider these uncertainties. '.

Specific comment 41 529 'dynamic' –> 'dynamics'

Response: 'dynamic' will be revised to 'dynamics'.

Specific comment 42 595 'low trophic levels consumers': do you mean first order consumers?

Response: Here we would prefer 'low trophic levels consumers', which includes not only first order consumers but also second order consumers. Dietary preferences of zooplankton may change with environmental conditions such as temperature (Boersma et al. 2016). For example, the copepod Temora longicornis preferred the cryptophyte Rhodomonas salina at higher temperatures, while it preferred the heterotrophic dinoflagellate Oxyrrhis marina at lower temperatures (Boersma et al. 2016). In the studies we cited (Garzke et al. 2016; Garzke et al. 2017), the influences of warming and ocean acidification were studied in a community of calanoid copepods, which showed feeding preferences between phytoplankton and microzooplankton. Thus, it is more precise to use the term 'low trophic levels consumers' here.

Specific comment 43 606 'relationship' –> 'relationships'

Response: 'relationship' will be revised to 'relationships'.

Specific comment 44 612 How does the temperature and CO2 relate to future ocean scenarios? That would be good to add to the introduction.

Response: Please see our reply to General comment I.

Specific comment 45 614 Wouldn't that contradict the argument you made in line 523-524 that these results cannot be scaled up to carbon export?

Response: The argument about carbon export will be revised. Please see our reply to Specific comment 40.

Specific comment 46 Table S2: the meaning of the column effect builder is not clear to me. What does main, two way and three way mean and how do these model outputs relate to the ones in table 2?

Response: In Table S2, 'main', 'two way' and 'three way' mean models containing first order effects of the three factors, second order interactions of all factors, and third order interactions of all factors, respectively. The selected models in Table 2 are shown in bold in Table S2. We will clarify the meaning of the column effect builder and the relationship between Table S2 and Table 2 in the revised manuscript.

Specific comment 47 Table 2: It is not clear to me what a significant intercept in these models mean? Furthermore, I'm missing interaction terms for some of the variables. I would change PIC (ug/ml) to PIC population yield (ug/ml) to make it easier to connect with the text.

Response: A significant intercept means that the regression curve (or in case of linear correlations: regression line) does not pass through the origin. Table 2 only shows the results of selected models. For some variables such as PON:POC, the model with only first order effects of the three factors was selected, because it can best predict targets. Thus, there were no interaction terms for the variable PON:POC. PIC (and POC) population yield will be used to easily connect with the text.

Specific comment 48 Figure 1: I'm missing the results for N:P supply in this figure.

Response: We will add the results for N:P supply ratio in Fig. 1.

Specific comment 49 Figure 2: I'm missing the results for CO2 in this figure.

Response: We will add the results for pCO2 in Fig. 2.

Specific comment 50 Table S4 seems to be the only results in which standard deviations instead of standard errors are reported. For consistency reasons I would opt for standard errors here.

[Figure]

Response: We will show standard errors in Table S4.

Specific comment 51 Fig S2 is missing the (mean +/- SE) from the legend. Or is standard deviation expressed here?

Response: Data in Fig. S2 are expressed as mean $\pm$ SE. As suggested, this information will be clarified in the revised manuscript.

References

Ahlgren, G. 1985. Growth of Oscillatoria agardhii in chemostat culture 3. Simultaneous limitation of nitrogen and phosphorus. Br. Phycol. J. 20: 249-261. doi: 10.1080/00071618500650261

Baek, S. H., S. Shimode, M.-S. Han, and T. Kikuchi. 2008. Growth of dinoflagellates, Ceratium furca and Ceratium fusus in Sagami Bay, Japan: The role of nutrients. Harmful Algae 7: 729-739

Bi, R., C. Arndt, and U. Sommer. 2012. Stoichiometric responses of phytoplankton species to the interactive effect of nutrient supply ratios and growth rates. J. Phycol. 48: 539-549. doi: 10.1111/j.1529-8817.2012.01163.x

Bi, R., S. M. H. Ismar, U. Sommer, and M. Zhao. 2017. Environmental dependence of the correlations between stoichiometric and fatty acid-based indicators of phytoplankton food quality. Limnol. Oceanogr. 62: 334-347. doi: 10.1002/lno.10429

Boersma, M., K. A. Mathew, B. Niehoff, K. L. Schoo, R. M. Franco-Santos, and C. L. Meunier. 2016. Temperature driven changes in the diet preference of omnivorous copepods: no more meat when it's hot? Ecol. Lett. 19: 45-53. doi: 10.1111/ele.12541

Bolker, B. M., M. E. Brooks, C. J. Clark, S. W. Geange, J. R. Poulsen, M. H. H. Stevens, and J.-S. S. White. 2009. Generalized linear mixed models: a practical guide for ecology and evolution. Trends Ecol. Evol. 24: 127-135. doi: 10.1016/j.tree.2008.10.008

Borchard, C., and A. Engel. 2012. Organic matter exudation by Emiliania huxleyi under

simulated future ocean conditions. Biogeosciences 9: 3405-3423. doi: 10.5194/bg-9-3405-2012

Boyd, P. W., R. Strzepek, F. Fu, and D. A. Hutchins. 2010. Environmental control of open-ocean phytoplankton groups: Now and in the future. Limnol. Oceanogr. 55: 1353-1376. doi: 10.4319/lo.2010.55.3.1353

Bracewell, S. A., E. L. Johnston, and G. F. Clark. 2017. Latitudinal variation in the competition-colonisation trade-off reveals rate-mediated mechanisms of coexistence. Ecol. Lett. 20: 947-957. doi: 10.1111/ele.12791

Cherif, M., and M. Loreau. 2010. Towards a more biologically realistic use of Droop's equations to model growth under multiple nutrient limitation. Oikos 119: 897-907. doi: 10.1111/j.1600-0706.2010.18397.x

De Bodt, C., N. Van Oostende, J. Harlay, K. Sabbe, and L. Chou. 2010. Individual and interacting effects of $pCO_2$ and temperature on Emiliania huxleyi calcification: study of the calcite production, the coccolith morphology and the coccosphere size. Biogeosciences 7: 1401-1412. doi: 10.5194/bg-7-1401-2010

Doney, S. C., and others. 2012. Climate change impacts on marine ecosystems. Annu. Rev. Mar. Sci. 4: 11-37. doi: 10.1146/annurev-marine-041911-111611

Feng, Y. 2015. Environmental controls on the physiology of the marine coccolithophore Emiliania huxleyi strain NIWA 1108. Ph.D. thesis. University of Otago.

Feng, Y., M. Y. Roleda, E. Armstrong, P. W. Boyd, and C. L. Hurd. 2017. Environmental controls on the growth, photosynthetic and calcification rates of a Southern Hemisphere strain of the coccolithophore Emiliania huxleyi. Limnol. Oceanogr. 62: 519-540. doi: 10.1002/lno.10442

Feng, Y., M. E. Warner, Y. Zhang, J. Sun, F.-X. Fu, J. M. Rose, and D. A. Hutchins. 2008. Interactive effects of increased $pCO_2$, temperature and irradiance on the marine coccolithophore Emiliania huxleyi (Prymnesiophyceae). Eur. J. Phycol. 43: 87-98. doi:

10.1080/09670260701664674

Frère, C. H., M. Kruetzen, J. Mann, R. C. Connor, L. Bejder, and W. B. Sherwin. 2010. Social and genetic interactions drive fitness variation in a free-living dolphin population. Proc. Natl. Acad. Sci. U. S. A. 107: 19949-19954. doi: 10.1073/pnas.1007997107

Fu, F.-X., E. Yu, N. S. Garcia, J. Gale, Y. Luo, E. A. Webb, and D. A. Hutchins. 2014. Differing responses of marine N2 fixers to warming and consequences for future diazotroph community structure. Aquat. Microb. Ecol. 72: 33-46. doi: 10.3354/ame01683

Garzke, J., T. Hansen, S. M. H. Ismar, and U. Sommer. 2016. Combined effects of ocean warming and acidification on copepod abundance, body size and fatty acid content. Plos One 11: e0155952. doi: 10.1371/journal.pone.0155952

Garzke, J., U. Sommer, and S. M. H. Ismar. 2017. Is the chemical composition of biomass the agent by which ocean acidification influences on zooplankton ecology? Aquat. Sci. doi: 10.1007/s00027-017-0532-5

Harada, N., and others. 2012. Enhancement of coccolithophorid blooms in the Bering Sea by recent environmental changes. Global Biogeochem. Cy. 26. doi: 10.1029/2011gb004177

IPCC. 2014. Climate change 2014: Synthesis report. Contribution of working groups I, II and III to the fifth assessment report of the intergovernmental panel on climate change, IPCC. Geneva, Switzerland.

Jamil, T., C. Kruk, and C. J. F. Ter Braak. 2014. A unimodal species response model relating traits to environment with application to phytoplankton communities. Plos One 9: e97583. doi: 10.1371/journal.pone.0097583

Joint, I., S. C. Doney, and D. M. Karl. 2011. Will ocean acidification affect marine microbes? Isme Journal 5: 1-7. doi: 10.1038/ismej.2010.79

Lafon, V., A. Martins, M. Figueiredo, M. A. Melo Rodrigues, I. Bashmachnikov, A. Mendonca, L. Macedo, and N. Goulart. 2004. Sea surface temperature distribution in the Azores region. Part I: AVHRR imagery and in situ data processing. Arquipelago Boletim da Universidade dos Acores Ciencias Biologicas e Marinhas: 1-18

Lewandowska, A. M., D. G. Boyce, M. Hofmann, B. Matthiessen, U. Sommer, and B. Worm. 2014. Effects of sea surface warming on marine plankton. Ecol. Lett. 17: 614-623. doi: 10.1111/ele.12265

Lynn, S. G., S. S. Kilham, D. A. Kreeger, and S. J. Interlandi. 2000. Effect of nutrient availability on the biochemical and elemental stoichiometry in the freshwater diatom Stephanodiscus minutulus (Bacillariophyceae). J. Phycol. 36: 510-522. doi: 10.1046/j.1529-8817.2000.98251.x

Matson, P. G., T. M. Ladd, E. R. Halewood, R. P. Sangodkar, B. F. Chmelka, and D. Iglesias-Rodriguez. 2016. Intraspecific differences in biogeochemical responses to thermal change in the coccolithophore Emiliania huxleyi. Plos One 11: e0162313. doi: 10.1371/journal.pone.0162313

Matthiessen, B., S. L. Eggers, and S. A. Krug. 2012. High nitrate to phosphorus regime attenuates negative effects of rising $pCO_2$ on total population carbon accumulation. Biogeosciences 9: 1195-1203. doi: 10.5194/bg-9-1195-2012

Nifong, R. L., M. J. Cohen, and W. P. Cropper, Jr. 2014. Homeostasis and nutrient limitation of benthic autotrophs in natural chemostats. Limnol. Oceanogr. 59: 2101-2111. doi: 10.4319/lo.2014.59.6.2101

Piepho, M., M. T. Arts, and A. Wacker. 2012. Species-specific variation in fatty acid concentrations of four phytoplankton species: does phosphorus supply influence the effect of light intensity or temperature? J. Phycol. 48: 64-73. doi: 10.1111/j.1529-8817.2011.01103.x

Rosas-Navarro, A., G. Langer, and P. Ziveri. 2016. Temperature affects the morphology and calcification of Emiliania huxleyi strains. Biogeosciences 13: 2913-2926. doi:

10.5194/bg-13-2913-2016

Rosenblatt, A. E., and O. J. Schmitz. 2016. Climate change, nutrition, and bottom-up and top-down food web processes. Trends Ecol. Evol. 31: 965-975. doi: 10.1016/j.tree.2016.09.009

Rost, B., and U. Riebesell. 2004. Coccolithophores and the biological pump: responses to environmental changes, p. 99-125. In H. R. Thierstein and J. R. Young [eds.], Coccolithophores: From molecular processes to global impact. Springer.

Schiettecatte, L. S., H. Thomas, Y. Bozec, and A. V. Borges. 2007. High temporal coverage of carbon dioxide measurements in the Southern Bight of the North Sea. Mar. Chem. 106: 161-173. doi: 10.1016/j.marchem.2007.01.001

Sett, S., L. T. Bach, K. G. Schulz, S. Koch-Klavsen, M. Lebrato, and U. Riebesell. 2014. Temperature modulates coccolithophorid sensitivity of growth, photosynthesis and calcification to increasing seawater $pCO_2$. PLoS ONE 9: e88308. doi: 10.1371/journal.pone.0088308

Skau, L. F. 2015. Effects of temperature and phosphorus on growth, stoichiometry and size in three haptophytes. M.S. thesis. University of Oslo.

Terry, K. L., J. Hirata, and E. A. Laws. 1985. Light-, nitrogen-, and phosphorus-limited growth of Phaeodactylum tricornutum Bohlin strain TFX-1: Chemical composition, carbon partitioning, and the diel periodicity of physiological processes. J. Exp. Mar. Biol. Ecol. 86: 85-100

Toseland, A., and others. 2013. The impact of temperature on marine phytoplankton resource allocation and metabolism. Nat. Clim. Change 3: 979-984. doi: 10.1038/nclimate1989

Ziveri, P., M. Passaro, A. Incarbona, M. Milazzo, R. Rodolfo-Metalpa, and J. M. Hall-Spencer. 2014. Decline in coccolithophore diversity and impact on coccolith morphogenesis along a natural $CO_2$ gradient. Biol. Bull. 226: 282-290

[Figure]

Please also note the supplement to this comment:
https://www.biogeosciences-discuss.net/bg-2017-162/bg-2017-162-AC1-supplement.pdf

---

## Author Response (AR1)

**Authors' response to referees: comments of the referees are in black, and responses are in blue.**

**Responses to comments from Reviewer 1**

**Comment 1**

Line 30: "PIC" for the first appearance, should be marked it's the abbreviation of "particulate inorganic carbon". Also for "POC".

Response:

As suggested, all abbreviations are spelled out once in full upon their first appearance in Abstract and the main manuscript text.

**Comment 2**

Line 31-32: "10:1, 24:1 and 63:1" are the ratios of N:P, the unite "mol mol-1" , not necessarily shown.

Response:

As suggested, we removed the unit 'mol mol$^{-1}$' and state 'molar ratios 10:1, 24:1 and 63:1'. (See Page 2, Line 31; Page 8, Line 155)

**Comment 3**

Line 87-92: "E.huxleyi is expanding its range poleward", why then gave an example of the subtropical area.

Response:

The reference to a study in the subtropical area was removed. A study in the Bering Sea is added, in which Harada et al. (2012) found that warming and freshening have promoted *Emiliania huxleyi* blooms since the late 1970s. (See Page 5, Lines 92-94)

**Comment 4**

L149-151: "The target values were chosen to reflect a present and future regime of each factor", however, the pCO2 concentrations 560 and 2400 µatm they used, can hardly be considered reasonable. An explanation why a gap in the CO2 concentrations was so big.

Response:

In plankton-rich waters, respiration plus atmospheric $CO_2$-enrichment can drive regional $pCO_2$ up to 900 $\mu$atm at times even today. Considerable seasonal, depth and regional variations of $pCO_2$ have been observed in the present-day ocean (Joint et al. 2011). For example, up to 900 $\mu$atm of $pCO_2$ was observed in August in the Southern Bight of the North Sea, with a lower $pCO_2$ (192 $\mu$atm) in April (Schiettecatte et al. 2007). A natural $pCO_2$ gradient of 292 to 8828 $\mu$atm was reported off Culcano Island, Italy (Ziveri et al. 2014). In the future oceans, $pCO_2$ will increase with rising atmospheric $CO_2$, being 851-1370 $\mu$atm by 2100 and 1371-2900 $\mu$atm by 2150 (RCP8.5 scenario of the IPCC report 2014) (IPCC 2014).

In the present study, the chosen values of $pCO_2$ cover the range of typical levels of $pCO_2$ in the present-day ocean and future ocean projections. Such a big gap in the value of $pCO_2$ was used to test the response of *E. huxleyi* to a considerable, yet realistic variation of $pCO_2$.

To clarify the reason of $pCO_2$ set-up in our study, a detailed explanation is added in the revised manuscript. (See Page 7, Lines 134-141)

**Comment 5**

Line 172: Can they write in detail about how "the specific growth rate of 20% of µmax was applied". I'm curious and puzzled about the reason and methods of how the 20% of µmax (µ) was realized. Usually, specific growth rate is not expressed by %.

Response:

Using % of $\mu_{max}$ guarantees that the strength on nutrient deficiency is equal through all temperature and $pCO_2$ treatments. A fixed value of µ would mean weak deficiency when $\mu_{max}$ is low, and strong deficiency when it is high. Based on the gross growth rate ($\mu$= 20% of $\mu_{max}$ (day$^{-1}$)), the equivalent daily renewal rate ($D$, day$^{-1}$) can be estimated according to the equation $D = 1- e^{-\mu t}$, where $t$ is renewal interval (day) (here $t = 1$ day). Thus, the volume of the daily renewal incubation water can be calculated by multiplying $D$ with the total volume of incubation water.

In the revised manuscript, we use the term 'gross growth rate' instead of specific growth rate. The term 'the gross growth rate' is explained on Page 9, Lines 184-185, which results from the process of reproduction alone. We also provide the detail about how 20% of $\mu_{max}$ was realized and applied and the reason of using % of $\mu_{max}$ in the revised manuscript. (See Page 9, Lines 185-191)

**Comment 6**

Line175-176: They said that the incubation water was exchanged with fresh seawater, since the culture medium was partially renewed according to the renewal rate D, the N:P ratios might deviate the target supply ratios in the remained medium due to differential consumption of N and P, can they give some information to show that the N:P supply ratios are stable after several rounds of renewal.

Response:

'fresh' seawater here implies freshly made seawater medium with the target N:P supply ratio, not only fresh seawater. Indeed, nutrient concentrations in semi-continuous culture may deviate from the target values due to consumption. Semi-continuous cultures, as a practical surrogate for fully continuous culture, have been successfully used to study the effect of nutrients on phytoplankton stoichiometry and fatty acid composition (Terry et al. 1985; Lynn et al. 2000; Piepho et al. 2012; Feng et al. 2017). While we did not measure the N and P in the media daily, semi-continuous cultures can be applied to study the effect of N:P supply ratio on *E. huxleyi* stoichiometry and fatty acid composition.

In the revised manuscript, we explain the successful usage of semi-continuous cultures in studies of phytoplankton stoichiometric and biochemical composition (See Pages 8, Lines 158-161). We also clarify that the incubation water was exchanged with freshly made seawater medium. (See Page 9, Line 192)

**Comment 7**

Line 178: It seems that the cell concentration was extremely high, the cell concentration range should be provided.

Response:

In our study, the final cell density at the steady state ranged between $1.50 \times 10^5 -$ $17.8 \times 10^5$ cells mL$^{-1}$, with the average value of $7.95 \times 10^5$ cells mL$^{-1}$. High cell density ($> 1 \times 10^6$ cells mL$^{-1}$) was observed in six out of 18 treatments. The average value of cell density in our study was consistent with the range of those in previous studies. For example, De Bodt et al. (2010) reported the maximum cell density of $6.84 \times 10^5$ cells mL$^{-1}$ when testing the effects of $pCO_2$ and temperature on *E. huxleyi* calcification.

As suggested, the range of cell densities is shown in the revised manuscript. (See Page 10, Lines 203-204)

**Comment 8**

Line 180: What do the authors mean by "the net growth rate (r)", what's the difference between r and μ? Confusing wordings or mis-understood definations?

Response:

To clarify the difference between $\mu$ and $r$, we now use the term 'gross growth rate' for $\mu$ (resulting from the process of reproduction alone), while $r$ (net growth rate) is the difference between the gross growth rate and the loss rate ($r = \mu - D$).

The difference between gross growth rate and net growth rate is clarified on Page 9, Lines 184-185 and 197-198.

**Comment 9**

Line 203: Here "was" should be "were".

Response:

As suggested, the word 'was' was replaced by 'were'. (See Page 11, Line 222)

**Comment 10**

Line 241: Is this theory applicable in all species and in any conditions.

Response:

This hypothesis was proposed by Cherif and Loreau (2010), suggesting that realized maximum growth rates (i.e., the observed maximum growth rate in the present study, $\mu_{max}$) should be equal for essential, non-substantial resources for phytoplankton species. This assumption was supported by both theoretical and empirical evidence, 1) lab experiments showed little or no luxury uptake of resources at the highest growth rate; 2) the maximum capacity of the uptake machinery should not be oversized for a given resource based on economical design (Cherif and Loreau 2010). Similar $\mu_{max}$ in different nutrient conditions has been observed for different phytoplankton species in empirical experiments (e.g., Ahlgren 1985; Baek et al. 2008; Bi et al. 2012). The model presented in Cherif and Loreau (2010) has also been successfully used to study how nutrient gradients influence stoichiometry of autotrophs in natural chemostats (Nifong et al. 2014).

In the present study, we had only one value of $\mu_{max}$ for each nutrient treatment under different temperature and $pCO_2$ conditions, thus the effect of N:P supply ratio cannot be tested with ANOVA efficiently. In the revised manuscript, we tested the response of $\mu_{max}$ to temperature, N:P supply ratio and $pCO_2$ using GLMMs. The results of GLMMs were consistent with those of ANOVA, showing a highly significant effect of temperature on $\mu_{max}$. As the chosen best model contained only first order effects, no significant interactions between the three environmental factors were detected. The non-significant response of $\mu_{max}$ to N:P supply ratio in *E. huxleyi* is consistent with the assumption of Cherif and Loreau (2010).

In the revised manuscript, methods and results of ANOVA were removed. We also revised the Results and Discussion sections according to the new results of GLMMs on $\mu_{max}$. (See Page 14, Lines 294-303; Page 19, Lines 399-403)

**Comment 11**

Line1103: Why there is no panel for the pCO2 effect in Fig.2.

Response:

The effect of $pCO_2$ on stoichiometric C:N:P is added in Fig. 3 in the revised manuscript.

**Comment 12**

Line 1112: As I read from the "experimental setup" part, this study investigates the combined effects of temperature, pCO2 and N:P supply ratios on E.huxleyi. Why in Fig 3. the combined effects of N:P supply ratio and pCO2 are not considered, i.e. pCO2 is not considered in panel (a), (b), (c), and N:P supply ratio is not considered in panel (d), (e) and (f). The same question for Fig. 4, 5 and 6.?

Response:

In the present study, we tested the effects of temperature, N:P supply ratio and $pCO_2$ on *E. huxleyi* using GLMMs. The selected best models contain only first order effects, or first order effects and second order interactions of the three factors, while models containing third order interactions of the three factors were not selected for any response variables (Table S2). Furthermore, significant interactions between temperature and N:P supply ratio, and between temperature and $pCO_2$ were detected for cellular PIC and POC contents, and the proportion of DHA. However, the significant interaction between N:P supply ratio and $pCO_2$ was only found for the proportion of SFAs. Please see Table 2 for a systematic summary. Thus, the interactions between temperature and N:P supply ratio, and between temperature and $pCO_2$ are shown in figures 1-4, while that between N:P supply ratio and $pCO_2$ is only shown for SFAs in the supporting information Fig. S2.

To clarify the information above, we took the following actions in the revised manuscript:

1) In the Methods, we explain more about the best models selected for different response variables. (See Page 13, Lines 272-278)

2) The selected best models and significant interactions are also briefly stated in figure legends in Fig. 1-4. (See Page 45, Lines 1059-1083)

**Responses to comments from Reviewer 2**

**General comment I**

My main concerns with the manuscript in its current form are the framing of the experimental manipulations to global patterns and the length of the discussion. There seems to be a mismatch between projected temperature and CO2 conditions in future oceans and the ones you manipulated. I would like to know how you would translate your results to a future ocean scenario as the CO2 concentrations in your lowest treatment are higher than currently measured in global oceans (max 440 ppm; Bakker et al. (2016)). On a similar note, how do the three temperature treatments with a difference of 12C relate to future ocean projections?

Response:

The chosen levels of $p$CO$_2$ and temperature in this study were set based on the reasons below:

1) $p$CO$_2$. Please see our reply to Reviewer 1, Comment 4. In plankton rich waters respiration plus atmospheric CO$_2$-enrichment can drive regional $p$CO$_2$ up to 900 $\mu$atm at times today. For example, up to 900 $\mu$atm of $p$CO$_2$ was observed in August in the Southern Bight of the North Sea (Schiettecatte et al. 2007). A much higher $p$CO$_2$ (a natural $p$CO$_2$ gradient of 292 to 8828 $\mu$atm) was observed off Culcano Island, Italy (Ziveri et al. 2014). In the future oceans, $p$CO$_2$ will increase with rising atmospheric CO$_2$, being 851-1370 $\mu$atm by 2100 and 1371-2900 $\mu$atm by 2150 (RCP8.5 scenario of the IPCC report 2014) (IPCC 2014). Therefore, the chosen values of $p$CO$_2$ in the present study cover the range of typical levels of $p$CO$_2$ in the present-day ocean and future ocean projections.

2) Temperature. Water surface temperatures at the Azores vary between ~12 to 29 ℃ (Lafon et al. 2004), with the inter-annual average temperature between 16 to 22 ℃ and peaks usually reaching a maximum of 24 to 25 ℃ (http://dive.visitazores.com/en/when-dive; last accessed date: 22.08.2017). Our temperature range setup was based on the study of Lewandowska et al. (2014), who chose a temperature increment of 6 ℃, according to the ocean general circulation model under the IPCC SRES A1F1 scenario. Annual mean sea surface temperature across the North Atlantic (0–60 °N) is projected to reach 29.8 °C in 2100 according to the ocean general circulation model (Lewandowska et al. 2014). We also chose this setup to compare with our previous results (Bi et al. 2017).

3) The ranges of $pCO_2$ and temperature in our study are identical in design with our previous work (Bi et al. 2017), which makes the comparison easier between results for different species.

The reasons for $pCO_2$ and temperature set-up are pointed out in the revised manuscript. (See Pages 6-7, Lines 128-141; Page 8, Lines 161-164)

**General comment II**

The discussion is quite lengthy and would benefit in my opinion to focus more on the interaction effects observed in the study as these are the core strength of the work and could advance the field. Perhaps you could reduce the amount of wording if you first discuss the solo effects and then go into all interaction effects in one paragraph (for C:N:P stoichiometry, PIC:POC separately). It seems that there is currently a lot of overlap in the things discussed in separate paragraphs.

Response:

As suggested, we first discuss the single effects and then continue to discuss interactive effects on C:N:P stoichiometry and PIC:POC. (See Pages 19-25, Lines 414-533)

**General comment III**

In addition, I'm missing the inclusion of the PON and POP contents underlying the responses in C:N:P stoichiometry, the results of N:P supply ratio on maximal growth rate and the (though non-significant) results of C:N:P stoichiometry in the different CO2 treatments (Fig. 2).

Response:

As suggested, the results of PON and POP contents, the N:P effect on $\mu_{max}$ in Fig. 1, and the $pCO_2$ effect on C:N:P stoichiometry in Fig. 2 and Fig. 3 are now shown in the revised manuscript.

**General comment IV**

From the introduction it is not clear that PIC and POC production will be discussed. In my opinion, the focus on C:N:P stoichiometry and underlying biochemical composition is the core of your work and introduced very well in the manuscript. I understand the importance of PIC:POC for calcifiers specifically, but I would advise to focus less on the PIC and POC contents, production rates and population yields and more on the C:N:P and fatty acids.

Response:

   As suggested, we removed the results of PIC and POC production rates and population yields in the section of Results, and moved the corresponding figures to the supporting information.

**General comment V**

A discussion on how changes in stoichiometry and fatty acids relate to each other would be a great addition to the discussion section.

Response:

   We add a graph (Fig. 5) to show how changes in stoichiometry and fatty acids relate to each other, i.e., the responses of PON:PUFAs and POP:PUFAs to temperature, N:P supply ratios and $p$CO$_2$. A section 'PON:PUFAs and POP:PUFAs' is added in Results (See Page 17, Lines 357-364). Accordingly, we discuss the implications of our results for ecology based on the relative changes in stoichiometry and fatty acids (See Pages 29-30, Lines 629-642).

**General comment VI**

What would be a great addition to the introduction are hypotheses on how temperature, CO2 and nutrient supply affect the C:N:P stoichiometry and fatty acid composition. Something similar to Figure 7, but then hypothetical. This would then furthermore help shape the discussion as you could refer back to these hypotheses.

Response:

   In the last paragraph of the introduction, we add hypotheses on how temperature, $p$CO$_2$ and nutrient supply affect elemental stoichiometry and fatty acid composition. (See Page 7, Lines 141-149)

**General comment V**II

A smaller comment, but the use of N:C and P:C ratios instead of C:N and C:P is not very commonly used in literature. The readability and comparison of these ratios to other studies would benefit greatly if they are expressed in C:N and C:P.

Response:

   We now use POC:PON and POC:POP, instead of N:C and P:C biomass ratios, in the revised manuscript.

**General comment VIII**

Furthermore, the reasoning behind the statistical methods used are not entirely clear for me. Some information in the method section on why this type of statistics are used and what the associated parameters mean would aid the reader in the understanding of the manuscript.

Response:

   Generalized linear mixed models (GLMMs) are appropriate for non-normal data such as counts or proportions, while classical statistical procedures such as ANOVA rely on normally distributed data (Bolker et al. 2009). GLMMs combine the properties of two statistical models (linear mixed models and generalized linear models) (Bolker et al. 2009) and have been widely used in ecology (e.g., Frère et al. 2010; Jamil et al. 2014; Bracewell et al. 2017), in which data sets are often non-normally distributed.

   We explain the reason why to choose GLMMs and what the associated parameter (link function) means in the revised manuscript. (See Page 12, Lines 248-254 and 259-264)

**Specific comment 1**

Do you mean community structure of phytoplankton?

Response:

   According to Doney et al. (2012), climate change may alter the physiological functioning, behavior, and demographic traits of organisms. These changes cascade from primary producers to upper trophic levels such as fish, seabirds and marine mammals. Therefore, community structure in the sentence in our manuscript means not only for phytoplankton but also for other trophic levels.

We hence clarify this as '---- community structure of different trophic levels ---'. (See Page 4, Line 73)

**Specific comment 2**

80-82 'via releasing CO2' is not really clear for me what this means and why coccolithophores are important components of the carbon cycle.

Response:

This sentence was revised to clarify that coccolithophores are not only important photosynthetic producers of organic matter (causing a draw-down of $CO_2$ in the surface layer), but also play predominant roles in the production and export of calcium carbonate to deeper layers (causing a net release of $CO_2$ to the atmosphere). (See Page 4, Lines 79-85)

**Specific comment 3**

What do you mean by a core feature? 'Element' – > 'elemental'

Response:

This sentence was revised to ' --- variability in *Emiliania huxleyi* C:N:P stoichiometry (cellular quotas and ratios of C, N and P) can also be important in ocean biogeochemistry.' (See Page 6, Lines 111-113)

'element' was changed to 'elemental'. (See Page 6, Line 113)

**Specific comment 4**

Food for which organism? Phytoplankton or zooplankton?

Response:

According to Rosenblatt and Schmitz (2016), shifts in resource nutrient content generally occur with shifts in consumer physiology and behavior, and they are often overlooked in studies of the responses of food web dynamics to climate change.

We thus clarify this sentence as ' --- shifts in resource nutrient content for consumers are often overlooked in climate change ecology --.'. (See Page 6, Line 118)

**Specific comment 5**

129-136 This comes as a surprise for me here and seems to fit better in the methodological section than in the introduction.

Response:

As suggested, these two sentences were moved to the methods section. (See Page 11, Lines 227-231; Pages 11-12, Lines 242-246)

**Specific comment 6**

PIC:POC is already a ratio

Response:

Wording was revised as PIC:POC throughout the text.

**Specific comment 7**

The manipulated $CO_2$ levels came as a surprise to me in the framework of current and future projections. Do you have specific reasons to choose these levels as I would have expected a lower 'ambient' $CO_2$ level (around 400 ppm)?

Response:

Please see our reply to General comment I.

**Specific comment 8**

Does the strain have a specific reference number (to make possible comparisons with other studies easier)?

Response:

As suggested, the specific reference number (internal culture collection reference code: A8) is now added. (See Page 8, Line 156)

**Specific comment 9**

does 'fresh' seawater imply that is was taken from sea at that day?

Response:

'fresh' seawater implies that freshly made seawater medium, but the seawater was not taken from the sea on that day. To clarify this, the sentence was revised as 'The incubation water was exchanged with freshly made seawater medium -----'. (See Page 9, Line 192)

**Specific comment 10**

Is there a specific reason why you choose GLMM's instead of the more classic ANOVA's?

Response:

Please see our reply to General comment VIII.

**Specific comment 11**

What are link functions?

Response:

The link function is a transformation of the target that allows estimation of the model (https://www.ibm.com/support/knowledgecenter/SSLVMB_21.0.0/com.ibm.spss.statistics.help/idh_glmm_target.htm; last accessed date: 14.08.2017). For example, identity link function is appropriate with any distribution except for multinomial, while logit can be used only with the binomial or multinomial distribution. We explain in the text what the link function is. (See Page 12, Lines 259-264)

**Specific comment 12**

222-239 This part of the statistics is quite difficult for me to follow. Could you explain a bit more about the different procedures and what they do?

Response:

In the revised manuscript, we took the following actions to explain more about GLMMs:

1) The reason why to choose GLMMs instead of classical statistical procedures is explained on Page 12, Lines 249-254.

2) We explain what link function is. (See Page 12, Lines 259-264)

3) According to differences in AICc values, the best model was selected for each response variable, which is now explained more on Page 13, Lines 272-278.

**Specific comment 13**

I would not assume mumax to be the same between nutrient treatments as that was the case in another study. Did you test this and was this the case in your study?

Response:

We tested the changes of $\mu_{max}$ between different nutrient treatments. Because there was only one value of $\mu_{max}$ in each nutrient treatment under different temperature and $pCO_2$ conditions, the effect of N:P supply ratio cannot be tested with ANOVA efficiently. In the revised manuscript, we show the results of GLMMs on the response of $\mu_{max}$ to temperature, N:P supply ratio and $pCO_2$ using GLMMs, which are consistent with those of ANOVA, showing a highly significant effect of temperature and non-significant effect of N:P supply ratio and $pCO_2$. As the chosen best model contained only first order effects, no significant interactions between the three environmental factors were detected.

In the literature, there are limited data on the response of $\mu_{max}$ in *E. huxleyi* to nutrient availability, while several studies reported the response of specific growth rate. According to Cherif and Loreau (2010), realized maximum growth rates (i.e., the observed maximum growth rate in the present study, $\mu_{max}$) should be equal for essential, non-substantial resources for phytoplankton species. This assumption was supported by both theoretical and empirical evidence, 1) lab experiments showed little or no luxury uptake of resources at the highest growth rate; 2) the maximum capacity of the uptake machinery should not be oversized for a given resource based on economical design (Cherif and Loreau 2010). For *E. huxleyi*, luxury consumptions for phosphate and nitrate are lower than other phytoplankton taxa (Rost and Riebesell

2004). Thus, the non-significant response of $\mu_{max}$ to N:P supply ratio in *E. huxleyi* in our study is consistent with the assumption of Cherif and Loreau (2010). Future work is suggested to study the response of $\mu_{max}$ in *E. huxleyi* under a wider range of nutrient conditions.

We revised the Results and Discussion sections according to the new results of GLMMs on $\mu_{max}$. (See Page 14, Lines 294-303; Page 19, Lines 399-403)

**Specific comment 14**

Is there a specific reason why you only used w2 for the mumax results and not for other results as for instance figure 7?

Response:

Error mean square cannot be obtained from GLMMs and thus $w^2$ cannot be calculated for response variables tested with GLMMs. In the revised manuscript, we show the response of $\mu_{max}$ using GLMMs. Thus, $w^2$ for $\mu_{max}$ was also removed.

**Specific comment 15**

Why would you use nested models when you have a full factorial design? In other words, what is the added value of these statistical tests? Can you relate your chosen temperatures to acclimatization of E.hux in your lab or the original population that was sampled? How are average annual water temperatures at the Azoren?

Response:

It is possible to use a nested model in a full-factorial design setting. The question a nested model addresses is that, whether one factor plays a role under one (or several) configuration(s) of another factor, but not under all configurations of that factor equally. The difference to e.g. a test including straight-forward interaction effects is that interaction terms describe systematic variation of one factor's effects over a gradient of the other, whereas a nested model can highlight if for example $p$CO$_2$ plays a role for fatty acid content only at intermediate temperature.

The added value of a nested model is explained on Page 13, Lines 281-283. Please see our reply to General comment I regarding average water surface temperatures.

The chosen temperature setup in our study is within the range of sea surface temperature at the Azores.

**Specific comment 16**

Did you determine the CO2 effects by post-hoc tests? As there was no overall effect of CO2 on maximum growth rate while you have a significant interaction effect, wouldn't that mean that the effect of temperature is dependent on the CO2 level, but not vice versa?

Response:

In our study, a post hoc test was applied only if there were significant effects in ANOVA. We thus did not determine the effect of $p\text{CO}_2$ by the post hoc test, as the effect of $p\text{CO}_2$ was not significant according to ANOVA.

We agree with the reviewer that the effect of temperature is dependent on the $\text{CO}_2$ level. In the revised manuscript, we used GLMMs to test the response of $\mu_{\max}$ (Please see our response to Specific comment 13). The results showed no significant interactions between temperature and $p\text{CO}_2$, while there was still a different trend of $\mu_{\max}$ to increase with increasing temperature between the two $p\text{CO}_2$ treatments.

We revised the results of $\mu_{\max}$ responses on Page 14, Lines 294-303. The discussion on $\mu_{\max}$ responses was also revised accordingly (See Page 19, Lines 399-403).

**Specific comment 17**

any particular reason to use N:C ratios as opposed to C:N ratios? The latter is used more often in literature and makes the comparison with the Redfield Ratio easier. For instance, a hump-shaped curve to temperature (or Ushaped curve, line 286) is also observed for a marine cyanobacterium (Fu et al. 2014). By having the ratios in N:C instead of C:N, comparison with other studies like these can get confusing. Furthermore, you did not report interaction effects of temperature and N:P supply ratio on N:P ratios. So how does the difference in temperature response under N and P deficiency (lines 287-288) relate to that?

Response:

We present the results of POC:PON and POC:POP in the revised manuscript to make the comparison with the Redfield Ratio and the results in the literature easier.

We found that, similar to the results in Fu et al. (2014), a hump-shaped curve to temperature was also observed for POC:PON in response to increasing temperature under N deficiency in our study.

Indeed, there was non-significant interaction between temperature and N:P supply ratio on PON:POP according to GLMMs. However, POC:PON responded significantly to temperature, showing a different trend of changes to increasing temperature under different N:P supply ratio. We thus present this nutrient-dependent response, as it need not be universal to constitute 'significant discovery'.

**Specific comment 18**

instead of biomass ratios, would it make sense to use PON:POP or POC:PON as that would already imply that it is biomass related. Related to that question, is the C:N ratio composed of TPC:PON or POC:PON? Furthermore, what is underlying the changes in stoichiometry? You have the results for POC content in Figure 4, but how do PON and POP change?

Response:

As suggested, we use POC:PON, POC:POP and PON:POP in the revised manuscript. The C:N biomass ratio is composed of POC:PON.

To explore what is underlying the changes in C:N:P stoichiometry, we analyzed the responses of cellular PON and POP contents. For example, a U-shaped curve was observed for the responses of cellular POC and PON contents to increasing temperature under N deficiency, which can explain the observed hump-shaped curve for the response of POC:PON. The detail results are shown on Pages 14-15, Lines 305-317, and discussed on Pages 20-21, Lines 429-435, 449-452 in the revised manuscript.

**Specific comment 19**

What is a PIC population yield?

Response:

A population yield of PIC is the PIC content per ml ($\mu g\ ml^{-1}$). This is now clarified in the revised manuscript. (See Page 25, Lines 540-541)

**Specific comment 20**

Technically, C:N:P is not a ratio but is composed of C:N and C:P ratios. Additionally, why did you chose to only highlight the N:P results?

Response:

As suggested, C:N:P biomass ratio was changed to C:N:P stoichiometry throughout the text.

The response of N:P biomass ratio is highlighted here because it had the highest percent changes among the three stoichiometric ratios. To clarify this, this sentence was revised as '----, showing a maximum of 62% changes under nutrient deficiency'. (See Page 17, Lines 369-370)

**Specific comment 21**

These interactions effects don't become clear from table 2, as there you only report the effects of the individual stressors.

Response:

Indeed, we did not observe significant effects of the three stressors on all response parameters, with significant interactive effects only observed for cellular POC content, and SFA and DHA proportions.

We clarify this information as '----, we also detected significant interactions between temperature, N:P supply ratios and $p$CO$_2$ on certain response variables (e.g., cellular POC content and DHA proportion) (Table 1),----'. (See Page 18, Lines 375-377)

**Specific comment 22**

'strains' instead of 'strain'

Response:

As suggested, the word was corrected to 'strains'. (See Page 18, Line 390)

**Specific comment 23**

It would be interesting to link this result with the origin of your strain. Does it fall in excepted patterns?

Response:

In our study, $\mu_{max}$ of *E. huxleyi* (from the Azores, ~ 38° N) was two to three times higher at the highest temperature than that at the lowest temperature, showing a similar change pattern with that in *E. huxleyi* (1.6 times higher at the higher temperature) from the Sargasso Sea (~20-35° N).

We add now the comparison between our results and the results in the literature in the revised manuscript. (See Page 18, Lines 394-396)

**Specific comment 24**

I could also argue it the other way, that the biogeographic origin of an E. huxleyi strain is important for their response to temperature. Like mentioned before, could you elaborate on this more?

Response:

We agree that the results show the importance of the biogeographic origin of an *E. huxleyi* strain for their response to temperature. We revised this sentence as 'The results above suggest that the biogeographic origin of an *E. huxleyi* strain is important for their response to temperature'. (See Pages 18-19, Lines 396-398)

**Specific comment 25**

Seems to contrast Table 1 and lines, were you show no effect of CO2 on maximal growth rate. Or is this based on post-hoc comparisons?

Response:

In the revised manuscript, GLMMs were used to test the response of $\mu_{max}$ to temperature, N:P supply ratio and $pCO_2$, showing non-significant interactions between the three factors. Thus, the discussion on the significant interactions between temperature and $pCO_2$ was removed in the section 'Responses of maximal growth rate' in the Discussion.

Please also see our reply to Specific comment 13 and 16.

**Specific comment 26**

387-389 Can you quantify these slopes as they come a bit as a surprise at this point in the manuscript.

Response:

These slopes were quantified. At the low $p$CO$_2$, the slopes were 0.13 and 0.026 at lower (12 and 18 ℃) and higher temperatures (18 and 24 ℃), respectively; at the high $p$CO$_2$, the slopes (0.04 – 0.06) were relatively constant. (See Page 19, Lines 399-403)

**Specific comment 27**

remove 'and'

Response:

'and' was removed. (See Page 19, Line 406)

**Specific comment 28**

If it is a conceptual graph you're referring to, I would be interested in the conceptual reasoning behind this response.

Response:

The conceptual reasoning behind is still unclear. The authors who proposed the conceptual graph suggested that one possible explanation is that increasing temperature may modulate the balance between a fertilizing effect of ocean carbonation and a metabolic repression by ocean acidification (Sett et al. 2014). This possible explanation is stated at the end of this paragraph. (See Page 19, Lines 408-412)

**Specific comment 29**

I would opt for 'C:N:P stoichiometry' instead of biomass ratios.

Response:

'C:N:P biomass ratios' was revised to 'C:N:P stoichiometry'. (See Page 19, Line 414)

**Specific comment 30**

What do you mean by 'prevailed the governing effect'?

Response:

Skau (2015) tested the effects of temperature and phosphorus on stoichiometry in three haptophytes, showing that phosphorus treatments had a stronger effect on C:P ratios in *E. huxleyi* compared to temperature.

We revised this sentence as '--- nutrient availability played a more important role than temperature (and $pCO_2$) for elemental stoichiometry ----'. (See Pages 19-20, Lines 418-419)

**Specific comment 31**

415-417 I really like Figure 7 as it gives a nice overview about your results. But what I'm missing there is the change in cellular N and P content. These results could help you in making conclusions about the changes in PON:POP, whether that is mainly due to N or P deficiency. Furthermore, should it be a table instead of a figure?

Response:

We add now the responses of cellular N and P contents in the revised manuscript. Please check our response to General comment III and Specific comment 18.

Fig. 7 is now shown as a table (Table 2) in the revised manuscript.

**Specific comment 32**

445-447 But given your result that the changes in N:C (or C:N) are stronger than those of P:C, what would be the mechanism behind that? Is there any current literature on that respect? Furthermore, if you bring in the argument of less P rich ribosomes with warming, wouldn't you have expected an decrease in P:C instead of the increase you observed?

Response:

In the literature, variable changes of POC:PON and POC:POP to warming were observed in *E. huxleyi*, showing positive (Borchard and Engel 2012), negative (Feng et al. 2008; Matson et al. 2016) and U-shaped responses (Rosas-Navarro et al. 2016). Similar to our study, Borchard and Engel (2012) also found a stronger change of

POC:PON than of POC:POP at higher P condition, while both biomass ratios increased with increasing temperature. The mechanism behind the stronger changes in POC:PON compared to POC:POP may be explained by the temperature-dependent physiology hypothesis, which shows that organisms in warmer conditions require fewer P-rich ribosomes, relative to N-rich proteins (Toseland et al. 2013). In our study, both POC:PON and POC:POP decreased with increasing temperature, while the change in POC:PON (8%) was larger than that in POC:POP (5%). Thus, the relative changes in POC:PON and POC:POP, as well as the increase in PON:POP, in response to increasing temperature in our study are consistent with the temperature-dependent physiology hypothesis (Toseland et al. 2013).

We revised this part to compare our results with current literature and to clarify the mechanisms (temperature-dependent physiology hypothesis) behind the changes of C:N:P stoichiometry in response to warming. (See Pages 21-22, Lines 452-465)

**Specific comment 33**

I'm missing here a coupling to your own experimental set-up, did you not find effects of CO2 on stoichiometry due to light conditions or nutrient loads? Can you compare your set-up with those from the studies you mentioned?

Response:

We add the comparison of experimental set-up between our study and previous work. For example, Feng et al. (2008) reported that rising $p\text{CO}_2$ caused the increase in POC:PON only at the high light condition (400 μmol photons $\cdot$ m$^{-2}$ $\cdot$ s$^{-1}$). The light intensity in our study (100 μmol photons $\cdot$ m$^{-2}$ $\cdot$ s$^{-1}$) was lower than that in Feng et al. (2008). In our study, we used relatively low light intensity (100 μmol photons $\cdot$ m$^{-2}$ $\cdot$ s$^{-1}$), did not investigate irradiance effects. Additional research is required to assess the effects of other important environmental factors such as irradiance and their interactions on C:N:P stoichiometry in our *E. huxleyi* strain. (See Page 22, Lines 474-481)

**Specific comment 34**

This is a rather fast transition for me from stoichiometry to cellular biomass. Perhaps this part fits better with the discussion paragraph on growth rates.

Response:

In this sentence, we discuss the responses of C:N:P stoichiometry and not cellular biomass. We revised this sentence as 'Taken together, our results indicate that C:N:P stoichiometry in *E. huxleyi* largely reflected the changes in N:P supply ratios, across different temperatures and $p$CO$_2$ levels.'. (See Page 22, Lines 482-484)

**Specific comment 35**

But you haven't looked at taxonomic composition as you study one species. As there is already such variability between strains and experiments with E.hux, I would shorten this paragraph and focus more on the drivers of variation in responses.

Response:

As suggested, the discussion in this paragraph was revised to focus more on the drivers of variation in stoichiometric responses: 'Taken together, our results indicate that C:N:P stoichiometry in *E. huxleyi* largely reflected the changes in N:P supply ratios, across different temperatures and $p$CO$_2$ levels. However, for two algal species from non-calcifying classes (the diatom *P. tricornutum* and the cryptophyte *Rhodomonas* sp.) temperature had the most consistent significant effect on stoichiometric ratios in our previous work (Bi et al. 2017). The results above are consistent with the ranking of environmental control factors in Boyd et al. (2010), which showed that temperature, nitrogen and phosphorus were ranked as important factors for major phytoplankton groups.'. (See Pages 22-23, Lines 482-489)

**Specific comment 36**

Refrain from starting a sentence with 'and'

Response:

'and' was removed from the beginning of the sentence. (See Page 24, Line 508)

**Specific comment 37**

This is vague for me, what other environmental drivers do you mean specifically?

Response:

According to previous studies, the interaction of $p$CO$_2$ with other environmental factors such as irradiance and temperature may be potential drivers on the changes in PIC:POC (Feng et al. 2008; De Bodt et al. 2010).

This sentence was removed as the whole paragraph was deleted.

**Specific comment 38**

'and the present study' should be within the brackets?

Response:

'the present study' is added within brackets. (See Page 24, Line 513)

**Specific comment 39**

CO2 would not be related to future oceans as the lowest treatment is already elevated.

Response:

Please see our reply to General comment I.

**Specific comment 40**

This argument is not clear to me and does not follow logically from your work. Yes, you have changes in PIC and POC yields with environmental changes, but why would that not scale up to carbon export?

Response:

We revised this sentence as 'It is worth noting that cellular PIC and POC contents are a measure for physiological response and cannot be directly used to infer population response, as different responses between cellular and population yields of PIC (and POC) (as µg ml$^{-1}$) to environmental changes were evident in previous work (Matthiessen et al. 2012) and the present study (Table S5, S6; Fig. S3, S4). Thus, scaling our results up to coccolithophores carbon export should consider these uncertainties.'. (See Page 25, Lines 538-543)

**Specific comment 41**

'dynamic' –> 'dynamics'

Response:

'dynamic' was revised to 'dynamics', while this sentence was removed in the revised manuscript.

**Specific comment 42**

'low trophic levels consumers': do you mean first order consumers?

Response:

Here we would prefer 'low trophic levels consumers', which includes not only first order consumers but also second order consumers. Dietary preferences of zooplankton may change with environmental conditions such as temperature (Boersma et al. 2016). For example, the copepod *Temora longicornis* preferred the cryptophyte *Rhodomonas salina* at higher temperatures, while it preferred the heterotrophic dinoflagellate *Oxyrrhis marina* at lower temperatures (Boersma et al. 2016). In the studies we cited (Garzke et al. 2016; Garzke et al. 2017), the influences of warming and ocean acidification were studied in a community of calanoid copepods, which showed feeding preferences between phytoplankton and microzooplankton. Thus, it is more precise to use the term 'low trophic levels consumers' here.

**Specific comment 43**

'relationship' –> 'relationships'

Response:

'relationship' was revised to 'relationships'. (See Page 28, Line 612)

**Specific comment 44**

How does the temperature and CO2 relate to future ocean scenarios? That would be good to add to the introduction.

Response:

Please see our reply to General comment I.

**Specific comment 45**

Wouldn't that contradict the argument you made in line 523-524 that these results cannot be scaled up to carbon export?

Response:

The argument about carbon export was revised. Please see our reply to Specific comment 40.

**Specific comment 46**

Table S2: the meaning of the column effect builder is not clear to me. What does main, two way and three way mean and how do these model outputs relate to the ones in table 2?

Response:

In Table S2, 'main', 'two way' and 'three way' mean models containing first order effects of the three factors, second order interactions of all factors, and third order interactions of all factors, respectively. The selected models in Table 1 are shown in bold in Table S2.

We clarify the meaning of the column effect builder and the relationship between Table S2 and Table 1 in the revised manuscript.

**Specific comment 47**

Table 2: It is not clear to me what a significant intercept in these models mean? Furthermore, I'm missing interaction terms for some of the variables. I would change PIC (ug/ml) to PIC population yield (ug/ml) to make it easier to connect with the text.

Response:

A significant intercept means that the regression curve (or in case of linear correlations: regression line) does not pass through the origin.

Table 1 only shows the results of selected models. For some variables such as POC:PON, the model with only first order effects of the three factors was selected, because it can best predict targets. Thus, there were no interaction terms for the variable POC:PON.

PIC (and POC) population yield is used in the revised manuscript. (See Table S5)

**Specific comment 48**

Figure 1: I'm missing the results for N:P supply in this figure.

Response:

The results for N:P supply ratio are shown in Fig. 1.

**Specific comment 49**

Figure 2: I'm missing the results for CO2 in this figure.

Response:

The results for $p$CO$_2$ are shown in Fig. 2 and Fig. 3.

**Specific comment 50**

Table S4 seems to be the only results in which standard deviations instead of standard errors are reported. For consistency reasons I would opt for standard errors here.

Response:

We show now standard errors in Table S4.

**Specific comment 51**

Fig S2 is missing the (mean +/- SE) from the legend. Or is standard deviation expressed here?

Response:

Data in Fig. S2 are expressed as mean $\pm$ SE. As suggested, this information is clarified in the revised manuscript.

[revised manuscript text omitted]

---

## Author Response (AR2)

**Authors' response to referees: comments of the referees are in black, and responses are in blue.**

**Responses to comments from Reviewer #2**

The authors did a good job revising the manuscript, as the current version is more concise and improved a lot. Furthermore, I think the presented results are extremely interesting and can further advance the field of climate change effects on stoichiometry and fatty acids composition. However, I still have some comments on the manuscript in its current form:

**General comment 1**

Table 2 at the moment includes both significant and non-significant changes, which can confuse the reader. I would therefore opt to either clearly indicate which changes are significant or alternatively only show significant changes in this table. This also relates to parts of the discussion (lines 449-465) where you discuss changes in C:P ratio with elevated temperature, which are actually non-significant in your analysis (table 1).

**Response:**

As suggested, we show only significant effects in Table 2. Also, in the Discussion, we only discuss the significant changes in POC:PON, as well as significant POC and PON content changes with warming, and state that POC:POP or PON:POP showed non-significant response to warming. (See Page 20, Lines 427-430)

**General comment 2**

I'm currently missing some lines in the results addressing the SFA responses to interactions between nutrient and CO2 treatment.

**Response:**

SFA responses to the interactions between N:P supply ratios and  $pCO_2$  are added in the Results. (See Page 16, Lines 341-348)

**General comment 3**

I don't fully understand how to interpret Figure 5, which I think came about after my previous comment on the relationship between changes in stoichiometry and fatty acid composition. My earlier comment was not directed towards the ratios between cellular contents and PUFAs, but towards a correlation between stoichiometric changes and fatty acid changes related to table 2. In other words, can changes in for instance PON (or POP, POC or PIC) content be related to changes in PUFAs (e.g. if PON goes up, does PUFAs also go up)? Thus, I would leave this figure out and discuss these putative correlations in the discussion.

**Response:**

As suggested, we now remove Fig. 5 and the section 3.4 in the Results. In our study, we observed an overall increase in POC:PON, POC:POP, and the proportions of PUFAs and DHA in *E. huxleyi* under future ocean scenarios (warming, N and P deficiency and enhanced  $pCO_2$ ) (Table 2), but a decrease in PUFA and DHA contents per biomass with enhanced  $pCO_2$  (Table S6). The relationship between changes in stoichiometry and FA composition in phytoplankton varies in a complex way with environmental conditions and algal taxonomy (Bi et al., 2014; Pedro Cañavate et al., 2017; Sterner and Schulz, 1998). Our findings thus indicate that elemental composition responses may be coupled with responses in essential FA composition in the strain of *E. huxleyi* studied under certain configurations of environmental drivers. We further discuss the implications of the changes in POC:PON, POC:POP and PUFAs for ecology. (See Pages 28-29, Lines 602-622)

**General comment 4**

Line 298 (and discussion lines 399-403): Why would you discuss an interaction effect that is not supported by the statistical analysis in table 1? I would omit this from the text.

**Response:**

In the revised manuscript, we now only discuss significant results. Therefore, we omit the results (on Page 14, Lines 398-303 in the previous version of the manuscript) and the discussion (on Page 19, Lines 399-412 in the previous version of the manuscript) on the interactive effect of temperature and  $pCO_2$  on  $\mu_{max}$ .

**General comment 5**

Line 502: I would not state that non-significant changes in POC content attributed to altered PIC:POC ratios.

**Response:**

This sentence is revised as 'The negative response of PIC:POC to rising  $pCO_2$  in our study was driven by the significant decrease in cellular PIC content (calcification), with cellular POC content (photosynthesis) showing non-significant changes (Table 1; Table 2)'. (See Page 22, Lines 476-477)

**Specific comment 1**

Line 43: 'compared to' instead of 'compared with'

**Response:**

As suggested, 'compared with' is changed to 'compared to'. (See Page 2, Line 43)

**Specific comment 2**

```
Line 117: '.' Instead of '; '
```

**Response:**

'.' is used instead of '; '. (See Page 6, Line 117)

**Specific comment 3**

Line 133 (and throughout the manuscript): typesetting of  $\,$   $\,$   $\,$

**Response:**

Typesetting of oC is done throughout the manuscript.

**Specific comment 4**

Line 139: As the IPCC is a model prediction, I would add some nuance to this sentence.

**Response:**

The sentence is revised as 'In future oceans,  $pCO_2$  is projected to increase with rising atmospheric CO2, ----'. (See Page 7, Lines 139)

**Specific comment 5**

Line 184-185: It is not clear to me what you mean with the gross growth rate. How does it only result from the process of reproduction? In these systems you would still have cell death as well right?

**Response:**

In the cultures of phytoplankton, there is negligible mortality due to the lack of predators. Therefore, 'gross growth rate ( $\mu$ )' in our culture systems means the rate of reproduction, while 'net growth rate' is used to describe the observed changes in abundance (i.e., the difference between the gross growth rate and the loss rate ( $r = \mu - D$ )). The definitions of gross growth rate and net growth rate above are referred to Lampert and Sommer (2007).

To clarify the definition of gross growth rate, we revised this sentence as 'The gross growth rate ( $\mu$  (d-1), resulting from the process of reproduction alone due to negligible mortality in cultures lacking predators (Lampert and Sommer, 2007)) was applied as 20% of  $\mu_{\text{max}}$ '. (See Page 9, Lines 185-186)

**Specific comment 6**

Line 312: 'values' instead of 'trend' as it otherwise suggests non-significance while it is supported by your analysis

**Response:**

The word 'values' is now used. (See Page 14, Line 308)

**Specific comment 7**

Line 351: 'N:P supply ratio' instead of 'nutrient deficiency' as in your experimental set-up one nutrient becomes deficient in replacement of the other over the experimental gradient. Related to that comment, in several parts of the discussion (lines 370 and 415) you write nutrient deficiency which I think should be phosphorus deficiency (as nitrogen is replete in these cases).

**Response:**

We now write 'N:P supply ratios' instead of 'nutrient deficiency' on Page 16, Lines 351-352. Also, we write 'P deficiency' instead of 'nutrient deficiency' in the Discussion. (See Page 17, Line 362; Page 18, Line 393)

**Specific comment 8**

Line 408: two times 'conceptual' in one sentence, consider revising.

**Response:**

Please check our response to General comment 4. As the discussion on the interactive effect of temperature and  $pCO_2$  on  $\mu_{max}$  is now removed, the sentence mentioned in this comment is also deleted.

**Specific comment 9**

Line 479: add 'and'

**Response:**

'and' was added on Page 21, Line 453.

**Specific comment 10**

Line 522: The example (txCO2) is inconsistent with the factors in the previous sentence (txnutrient)

**Response:**

For cellular particulate carbon contents, we observed significant interactions between temperature and N:P supply ratios, and between temperature and  $pCO_2$ (Table 2). Thus, the sentence is revised as 'Significant interactions were observed between temperature and N:P supply ratios, and between temperature and  $pCO_2$  on cellular particulate carbon contents in our study (Table 2)'. (See Page 23, Lines 496-498)

**Specific comment 11**

Line 548: 'MUFAs' instead of 'MUAFs'

Response:

The word is now corrected to 'MUFAs'. (See Page 24, Line 522)

**Specific comment 12**

Line 588: 'while' instead of 'with'

**Response:**

The word 'while' is used instead of 'with'. (See Page 26, Line 562)

**Specific comment 13**

Line 616: rephrase as no effect of N:P supply ratios on C:P, nor on PON and POP were observed.

**Response:**

The sentence is rephrased as 'We observed an overall increase in POC:PON (with warming and N deficiency) and POC:POP (with N and P deficiency) in *E. huxleyi*, while enhanced  $pCO_2$  showed no clear effects (Table 2)'. (See Page 27, Lines 588-590)

[revised manuscript text omitted]

0.111 | 0.024 0.912 |
| POC:PON (mol mol -1 )           | -                  | $2.741 \pm 0.081$       | 33.823         | <0.912
<0.001   |
| FOC.FON (mor mor )                         | Intercept
T     | $-0.008 \pm 0.004$      | -2.169         | <0.001
0.035    |
|                                            |                    |                         |                |                    |
|                                            | $pCO_2$            | <0.001 ±<0.001          | 0.153          | 0.879              |
| $\mathbf{POC}(\mathbf{POP}(m+1,m+1^{-1}))$ | N:P                | $-0.004 \pm 0.001$      | -5.430         | < 0.001            |
| POC:POP (mol mol -1 )           | Intercept          | $5.423 \pm 0.128$       | 42.300         | <0.001             |
|                                            | T                  | $-0.007 \pm 0.006$      | -1.242         | 0.220              |
|                                            | $pCO_2$            | <0.001 ±<0.001          | 0.069          | 0.945              |
| DON DOD ( 1 1-1)                           | N:P                | $0.012 \pm 0.001$       | 9.617          | <0.001             |
| PON:POP (mol mol -1 )           | Intercept          | $2.702 \pm 0.145$       | 18.590         | < 0.001            |
|                                            | Т                  | $0.001 \pm 0.007$       | 0.157          | 0.876              |
|                                            | $pCO_2$            | <0.001 ±<0.001          | -0.169         | 0.866              |
|                                            | N:P                | $0.016 \pm 0.001$       | 11.200         | <0.001             |
| PIC:POC                                    | Intercept          | $0.460 \pm 0.066$       | 7.010          | < 0.001            |
|                                            | Т                  | $0.025 \pm 0.003$       | 8.184          | <0.001             |
|                                            | $pCO_2$            | $<0.001 \pm <0.001$     | -12.837        | <0.001             |
|                                            | N:P                | <0.001 ±0.001           | -0.166         | 0.869              |
| SFA proportion (% of TFAs)                 | Intercept          | $3.506 \pm 0.145$       | 24.178         | < 0.001            |
|                                            | Т                  | $-0.012 \pm 0.008$      | -1.538         | 0.131              |
|                                            | $pCO_2$            | $<0.001 \pm <0.001$     | -0.238         | 0.813              |
|                                            | N:P                | $-0.004 \pm 0.003$      | -1.248         | 0.218              |
|                                            | $T \times pCO_2$   | $<\!0.001 \pm <\!0.001$ | 1.816          | 0.076              |
|                                            | $T \times N:P$     | $<\!0.001 \pm <\!0.001$ | 1.657          | 0.104              |
|                                            | $pCO_2 \times N:P$ | $<\!0.001 \pm <\!0.001$ | -2.487         | 0.016              |
| MUFA proportion (% of TFAs)                | Intercept          | $30.259 \pm 1.344$      | 22.518         | < 0.001            |
|                                            | Т                  | $-0.579 \pm 0.063$      | -9.240         | <0.001             |
|                                            | $pCO_2$            | $0.001 \pm < 0.001$     | 2.269          | 0.028              |
|                                            | N:P                | $-0.014 \pm 0.014$      | -1.050         | 0.299              |
| PUFA proportion (% of TFAs)                | Intercept          | $32.264 \pm 2.300$      | 14.028         | < 0.001            |
|                                            | Т                  | $0.638 \pm 0.107$       | 5.949          | <0.001             |
|                                            | $pCO_2$            | $-0.002 \pm 0.001$      | -2.769         | 0.008              |
|                                            | N:P                | $0.034\ \pm 0.023$      | 1.453          | 0.152              |
| DHA proportion (% of TFAs)                 | Intercept          | $2.204 \pm 0.185$       | 11.887         | < 0.001            |
|                                            | Т                  | $0.054 \pm 0.010$       | 5.611          | <0.001             |
|                                            | $pCO_2$            | $<\!0.001 \pm <\!0.001$ | 1.874          | 0.067              |
|                                            | N:P                | $0.010 \pm 0.004$       | 2.735          | 0.009              |
|                                            | $T \times pCO_2$   | $<\!0.001 \pm <\!0.001$ | -2.946         | 0.005              |
|                                            | $T \times N:P$     | $-0.001 \pm < 0.001$    | -2.898         | 0.006              |
|                                            | $pCO_2 \times N:P$ | $<\!0.001 \pm <\!0.001$ | 1.249          | 0.218              |
|                                            |                    |                         |                |                    |

Table 2. The changes in cellular elemental contents (as pg cell-1), elemental molar ratios and the proportions of major fatty acid groups and docosahexaenoic acid (DHA) (as % of total fatty acids) in response to warming, N and P deficiency and enhanced  $pCO_2$  in *Emiliania huxleyi*. Here, only significant changes are shown based on GLMM results in Table 1. Red and blue arrows indicate a mean percent increase and decrease in a given response, respectively.

**Fig. 4**

---

## Author Response (AR3)

**Authors' response to referees: comments of the referees are in black, and responses are in blue.**

**Responses to comments from Reviewer #2**

The authors did a good job revising the manuscript and accurately addressed my previous comments on the manuscript. I only have a few grammatical comments:

**Comment 1 and comment 2**

'the global carbon cycle'

'the global carbon cycle' or 'global carbon cycling'

Response:

As suggested, the word 'the' is added before 'global carbon cycle'. (See Pages 4 and 5, Lines 85 and 109)

**Comment 3**

'of' instead of 'on'

Response:

The word 'of' is used instead of 'on'. (See Page 9, Line 187)

**Comment 4**

missing 'a' before non-significant

Response:

'a' is added before 'non-significant' in several sentences. (See Page 15, Lines 324 and 330; Page 18, Line 395; Page 21, Line 448; Page 22, Lines 473)

**Comment 5**

'conditions' instead of 'condition'

Response:

The word is corrected to 'conditions'. (See Page 20, Line 437)

**The list of other changes to the manuscript**

Besides adjustments requested by the Reviewer, the following changes to the last version of the manuscript are also highlighted in blue.

1. Institutional affiliation was revised a bit. (See the title page)

2. The order of two grant numbers (41521064 and 41630966) was exchanged in Acknowledgements.

3. All significant intercepts are shown in bold in Table 1 and Table S5.

[revised manuscript text omitted]